# FSC-certified forest management benefits large mammals compared to non-FSC

Joeri A. Zwerts[1,2 ✉], E. H. M. Sterck[2,3], Pita A. Verweij[4], Fiona Maisels[5,6], Jaap van der Waarde[7], Emma A. M. Geelen[2], Georges Belmond Tchoumba[7], Hermann Frankie Donfouet Zebaze[7] & Marijke van Kuijk[1]

More than a quarter of the world's tropical forests are exploited for timber[1]. Logging impacts biodiversity in these ecosystems, primarily through the creation of forest roads that facilitate hunting for wildlife over extensive areas. Forest management certification schemes such as the Forest Stewardship Council (FSC) are expected to mitigate impacts on biodiversity, but so far very little is known about the effectiveness of FSC certification because of research design challenges, predominantly limited sample sizes[2,3]. Here we provide this evidence by using 1.3 million camera-trap photos of 55 mammal species in 14 logging concessions in western equatorial Africa. We observed higher mammal encounter rates in FSC-certified than in non-FSC logging concessions. The effect was most pronounced for species weighing more than 10 kg and for species of high conservation priority such as the critically endangered forest elephant and western lowland gorilla. Across the whole mammal community, non-FSC concessions contained proportionally more rodents and other small species than did FSC-certified concessions. The first priority for species protection should be to maintain unlogged forests with effective law enforcement, but for logged forests our findings provide convincing data that FSC-certified forest management is less damaging to the mammal community than is non-FSC forest management. This study provides strong evidence that FSC-certified forest management or equivalently stringent requirements and controlling mechanisms should become the norm for timber extraction to avoid half-empty forests dominated by rodents and other small species.

Commercial logging concessions cover more than one-quarter of the world's remaining tropical forests[1]. Forest certification schemes aim to have more positive socio-economic and environmental outcomes compared to conventional logging schemes. For example, the Forest Stewardship Council (FSC) aims to reduce direct environmental impacts by various means that include maintaining high conservation value forests and applying reduced impact logging practices (Supplementary Tables 1 and 2). A major concern for biodiversity is that timber extraction—by the creation of roads—opens previously remote forests, enabling illegal and unsustainable hunting[4–7]. This indirect effect of logging is known to mainly influence medium- to large-sized forest mammals, which are particularly vulnerable to human pressure[8]. FSC certification may alleviate these pressures because, among other measures, companies reduce accessibility to concessions by closing off old logging roads, prohibit wild meat transport and hunting materials, provide access to alternative meat sources for workers and their families, and carry out surveillance by rangers. An FSC certificate is valid for 5 years and logging companies are audited for compliance through third-party annual surveillance assessments.

In African tropical forests, FSC certification has been shown to be associated with reduced deforestation[9], improved working and living conditions of employees and benefit-sharing with neighbouring institutions[10]. Studies in Latin America suggest that mammal occupancy in FSC-certified sites is comparable to that of protected areas[11,12]. There is, however, little data on the status of faunal communities in FSC-certified versus non-FSC forests[2,3]. Most studies on the effectiveness of FSC certification for wildlife conservation have focused on one or a few sites or species at a time[13–16]. Although these studies reported a positive impact of FSC certification on wildlife compared to non-FSC concessions, their research designs did not account for explanatory variables such as concession location, land-use history or stochastic effects[17,18]. One study included several sites and species and found no effect of FSC certification[19]. However, that study investigated only bird species richness: bird dispersal distances are much higher than those of terrestrial mammals and may thus be a weak indicator of local management. In addition to simply comparing species diversity, it is important to compare population sizes between forest management types. Hunting does not necessarily completely extirpate wildlife species, especially when forests are connected, but rather results in population declines[4].

We used camera traps to assess whether FSC certification can mitigate the negative effects of timber extraction on wildlife by studying the encounter rate of a broad range of mammal species across several

[1]Ecology and Biodiversity, Utrecht University, Utrecht, The Netherlands. [2]Animal Behaviour & Cognition, Utrecht University, Utrecht, The Netherlands. [3]Animal Science Department, Biomedical Primate Research Centre, Rijswijk, The Netherlands. [4]Copernicus Institute of Sustainable Development, Utrecht University, Utrecht, The Netherlands. [5]Faculty of Natural Sciences, University of Stirling, Stirling, UK. [6]Wildlife Conservation Society, Global Conservation Program, New York, NY, USA. [7]WWF Cameroon, Yaoundé, Cameroon. ✉e-mail: j.a.zwerts@uu.nl

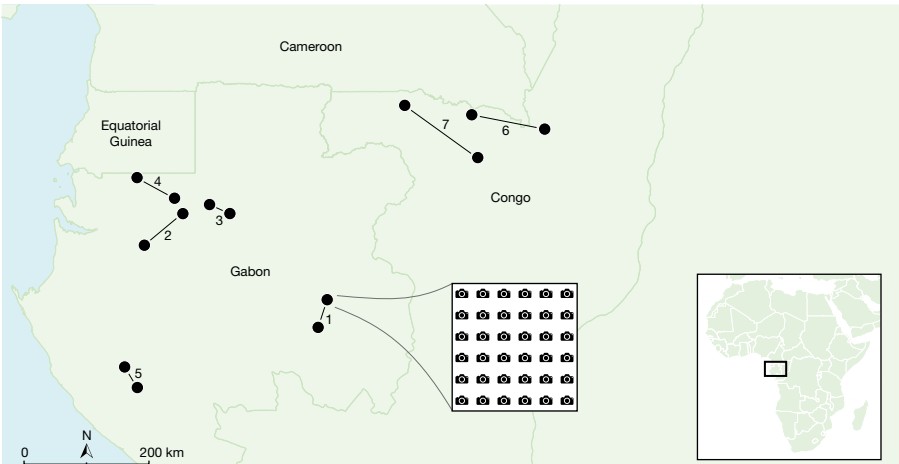

**Fig. 1 | Locations of the 14 paired logging concessions in Gabon and the Republic of Congo.** Between 28 to 36 cameras were deployed in each concession in systematic, 1 km spaced grids. Numbers and lines indicate the pairs of FSC-certified and non-FSC concessions.

sites. We compared small- to large-sized mammal observations across seven paired FSC-certified and non-FSC concessions in Gabon and the Republic of Congo (Fig. 1). Gabon and the Republic of Congo lie in western equatorial Africa (WEA). We included all companies that were FSC-certified between 2018 and 2021 in this region, except for one which refused to allow access. WEA is particularly suitable for these analyses, as its forests are reasonably intact and logging concessions are embedded in a matrix of contiguous forest, which are therefore mostly devoid of influences other than the effects of logging and hunting[20]. Wild meat hunting is pervasive throughout WEA, whereby logging increases hunting pressure by increasing access (logging roads) and through the arrival of people working in the concessions in once-remote forests[8]. By ensuring spatial pairing of the FSC-certified and non-FSC concessions we minimized the influence of regional landscape heterogeneity. We calculated mammal encounter rates and grouped mammal species into five body mass classes. The relative encounter rate of these classes could be used as a proxy for hunting pressure, as larger-bodied species are targeted more by hunters[8]. In addition, larger-bodied species recover more slowly from hunting compared to smaller-bodied species, resulting in lower abundances of large versus small species under higher hunting pressure[21,22]. Finally, we explored how FSC-certified forest management affects mammal encounter rate by taxonomic group and by IUCN Red List categories. We hypothesized that FSC certification would effectively decrease hunting pressure and therefore predicted a higher encounter rate of larger-bodied species in FSC-certified compared with non-FSC logging concessions.

We collected and catalogued nearly 1.3 million photos from 474 camera-trap locations for a total of 35,546 days, averaging 2,539 camera-trap days per concession (Extended Data Table 1). We detected a total of 55 mammal species (Extended Data Table 2). The mammal encounter rate estimated by our model (Fig. 2a) was 1.5 times higher in FSC-certified concessions compared to non-FSC concessions (Extended Data Table 3). We also found fewer signs of hunting (Fig. 2b) in FSC-certified than in non-FSC concessions. Estimated total faunal biomass derived from mammal encounter rates was 4.5 times higher in FSC-certified compared to non-FSC concessions (Extended Data Fig. 1). Larger species contributed more to the total biomass. We observed comparable species diversity in the two concession types, as only a few species, all with very low encounter rates, were lacking completely in one or other of the concession types (Extended Data Table 2).

The differences between mammal encounter rates in FSC-certified and non-FSC concessions increased with body mass (Fig. 3 and Extended Data Tables 3 and 4). FSC-certified concessions had higher encounter rates of mammals above 10 kg than non-FSC concessions but there was no difference for mammals below 10 kg. Model estimates showed that mammals in body mass classes over 100, 30–100 and 10–30 kg, had encounter rates that were 2.7, 2.5 and 3.5 times higher, respectively, in FSC-certified concessions compared to non-FSC concessions. Mammal encounter rates of the IUCN Red List categories critically endangered, near threatened and least concern were 2.7, 2.3 and 1.4 times higher, respectively, in FSC-certified compared to non-FSC concessions (Fig. 4 and Extended Data Tables 3 and 4).

Mammal encounter rate in FSC-certified and non-FSC concessions varied between taxonomic groups (Fig. 5 and Extended Data Tables 3 and 4). In FSC-certified concessions, forest elephants were encountered 2.5 times, primates 1.8 times, even-toed ungulates 2 times and

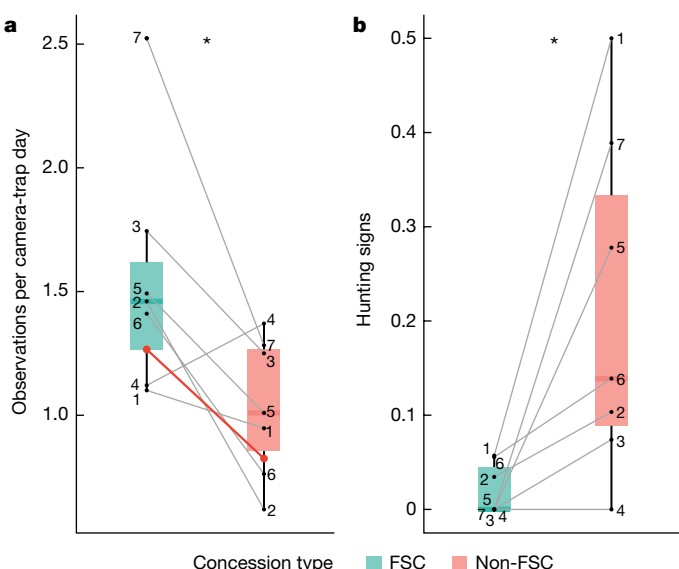

**Fig. 2 | Mammal encounter rate and hunting signs in FSC-certified and non-FSC concessions. a,b,** Encounter rate of all observed mammals ($P = 0.041$) (**a**) and proportion of camera locations with hunting signs ($P = 0.036$) (**b**). Numbers represent paired FSC-certified ($n = 7$) and non-FSC ($n = 7$) concessions. The red line in **a** represents the linear mixed model predicted fixed effect (certification status) and grey lines represent random effects (concession pairs). Differences between hunting signs in **b** were analysed using a two-sided Wilcoxon signed-rank test. Data are represented as boxplots, where central lines represent medians and lower and upper lines correspond to the first and third quartiles, whiskers reflect 1.5 times the interquartile range. *$P < 0.05$.

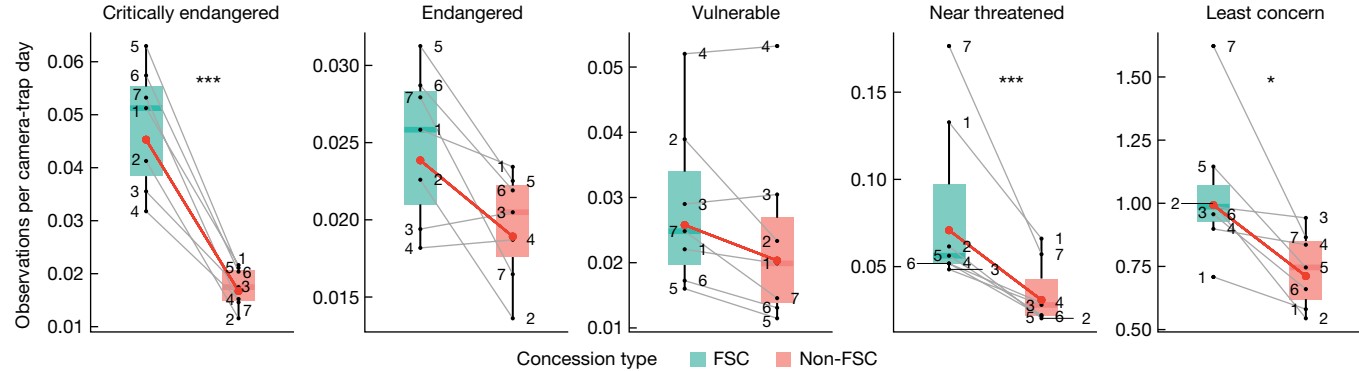

**Fig. 3 | Mammal encounter rate across five body mass classes in paired FSC-certified and non-FSC concessions.** Numbers represent paired FSC-certified (*n* = 7) and non-FSC (*n* = 7) concessions, red lines represent linear mixed model predicted fixed effects (certification status) and grey lines represent random effects (concession pairs). Data are represented as boxplots, where central lines represent medians and lower and upper lines correspond to the first and third quartiles, whiskers reflect 1.5 times the interquartile range. Pairwise comparisons were multivariate *t* adjusted. ***$P < 0.001$. Exact $P$ values are summarized in Extended Data Table 3. Note that the scales of the *y* axes vary. Silhouettes of *Gorilla gorilla, Syncerus caffer, Potamochoerus porcus, Cephalophus* sp., *Hyemoschus aquaticus, Philantomba monticola, Atherurus africanus* and mice were created by T. Markus.

carnivores 1.5 times more compared to non-FSC concessions. The encounter rate of pangolins and rodents did not differ.

## Discussion
### The loss of large mammals
We conducted a large-scale quantitative study to assess the impact of FSC-certified forest management on mammal encounter rate across

several logging concessions and for a broad range of mammals. Our data provide strong evidence that FSC-certified forest management results in higher overall mammal abundance, as approximated by encounter rate and faunal biomass relative to non-FSC forest management. This effect was most pronounced for species larger than 10 kg, which was consistent for all FSC–non-FSC concession pairs, probably because these medium to large species recover more slowly from population losses and may be targeted more often by hunters[21,22].

**Fig. 4 | Mammal encounter rate across IUCN Red List categories in paired FSC-certified and non-FSC concessions.** Numbers represent paired FSC-certified (*n* = 7) and non-FSC (*n* = 7) concessions, red lines represent linear mixed model predicted fixed effects (certification status) and grey lines represent random effects (concession pairs). Data are represented as boxplots, where central lines represent medians and lower and upper lines correspond to the first and third quartiles, whiskers reflect 1.5 times the interquartile range. Pairwise comparisons were multivariate *t* adjusted. ***$P < 0.001$, *$P < 0.05$. Exact $P$ values are summarized in Extended Data Table 3. Note that the scales of the *y* axes vary.

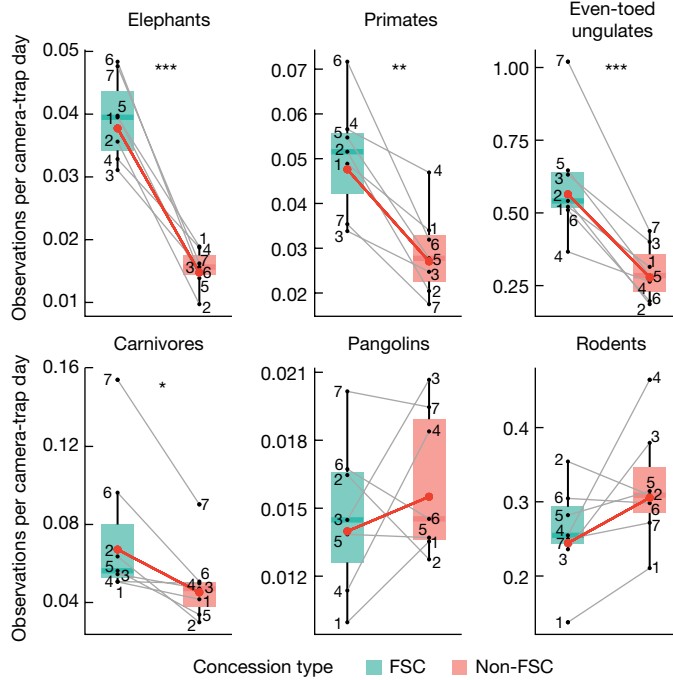

**Fig. 5 | Mammal encounter rate across six taxonomic groups in paired FSC-certified and non-FSC concessions.** Numbers represent paired FSC-certified ($n = 7$) and non-FSC ($n = 7$) concessions, red lines represent linear mixed model predicted fixed effects (certification status) and grey lines represent random effects (concession pairs). Data are represented as boxplots, where central lines represent medians and lower and upper lines correspond to the first and third quartiles, whiskers reflect 1.5 times the interquartile range. Pairwise comparisons were multivariate $t$ adjusted. ***$P < 0.001$, **$P < 0.01$, *$P < 0.05$. Exact $P$ values are summarized in Extended Data Table 3. Note that the scales of the $y$ axes vary.

Not all large species with reduced encounter rates may be commonly targeted for hunting but they are often indiscriminately affected by snaring[23]. Non-FSC concessions contained proportionally more rodents and other small species than did FSC-certified concessions (Extended Data Table 2). The lack of hunting impacts on small mammal populations suggests some form of density compensation is in place: the hunting pressure on small mammal populations might be compensated by higher reproductive rates and/or a release from competition and predation in the non-FSC concessions[24,25].

A particularly strong effect of FSC certification was found for the critically endangered forest elephant, which is in line with previous findings[14]. The distribution of this species is driven almost entirely by human activity: they avoid areas that are unsafe to them[26,27]. Their large home ranges can span several concessions[28], thus they may actively seek to reside not only in protected areas but also in FSC-certified concessions where measures to prevent illegal hunting are in place. This suggests that FSC-certified concessions may provide an important refuge for wide-ranging elephants. By contrast, no difference was found in pangolin encounter rate (they are among the most trafficked mammals[29]) between the two types of logging regimes. Two out of the three pangolin species present in WEA are relatively small and generally have higher reproduction rates than mammals in larger size classes. Moreovjer, all three pangolin species had low encounter rates in our study (Extended Data Table 2), probably because two pangolin species are semi-arboreal and are therefore not effectively captured by ground-based camera traps, which reduces our ability to draw strong conclusions about these species and warrants further research. We did not observe a loss of species that were encountered frequently in either FSC-certified or non-FSC concessions, nor did we expect to. This is because human population density in WEA is relatively low and the forests are still highly connected[20].

Conservation of large mammals through FSC certification brings wider benefits to forests, as these mammals play a pivotal role in ecological processes, including seed dispersal, seed predation, browsing, trampling, plant competition, nutrient cycling and predator–prey interactions[30]. It has also been suggested that forest carbon storage is higher when large mammal assemblages are more intact because the ecological processes they are part of (such as seed dispersal) often benefit large, high wood density trees[31–33] and the benefits of their conservation may far outweigh the cost[34]. Futhermore, by reducing the amount of wild meat available for human consumption, FSC-certified concessions or similar stringent schemes may also reduce the chance of zoonotic disease transmission[35].

## Methodological considerations

The FSC takes a comprehensive and all-encompassing perspective when it comes to managing and promoting sustainable forest management practices. This approach recognizes that forests are complex ecosystems with intricate interconnections between their various components, including flora, fauna, soil, water and climate. In logged tropical forests, controlling hunting is probably the most important factor for the reduction of environmental impacts[7]. We found more hunting signs in non-FSC concessions, which supports the interpretation that FSC effectively reduces hunting pressure, although counting hunting signs is likely to be a relatively weak measure of the quantification of hunting pressure[36]. Hunting has long been known to be the most important driver of forest fauna decline in central African logged forests[6,37] and the same phenomenon has been shown in Asia[7]. Of course, other factors such as retaining high conservation value areas and reduced impact logging practices are likely to contribute to the observed effects as well[38]. Our data do not allow for causal inference of the association of any of the specific measures implemented by FSC companies with the observed effects, as that would require setting up more detailed measure-based experiments.

For the sections of the concessions that we sampled, we ensured comparability between paired concessions. We maximized the similarity in geographic covariates that may drive variation in mammal abundance—elevation and distances to roads, rivers, human settlements and protected areas—between each pair of FSC-certified and non-FSC concessions (Extended Data Fig. 2 and Extended Data Table 5). Although we believe that these covariates are important drivers of mammal abundance[39], including these covariates did not greatly improve the models, which underscores that camera grid locations were sufficiently similar in terms of these confounding influences. Precise logging intensity and logging history data per camera were not available for most concessions because the planning schemes of companies and actual exploitation of cutting blocks often did not match. Slight differences in logging history are not expected to have a large effect on the data because mammals are mobile and can return quickly to areas that have been exploited[40]. Fourteen logging concessions may be a large sample size for tropical ecology studies[17] but a low sample size from a statistical perspective. Nonetheless, despite the small number of replicates, we found clear and consistent differences in encounter rate between FSC-certified and non-FSC forests.

We used encounter rate, defined as the number of observations divided by the number of camera-trap days. Encounter rates may be affected by unaccounted influences on detection probabilities[41], which may complicate comparisons between species or between sites. We compare individual species across management types, which renders differences in detection across species less relevant. For camera-trap sites, however, variation in visibility or other factors may affect the number of detections, even though mammal population sizes are similar. However, we found no differences in any relevant site covariates between treatments at the camera-trap level. Visibility at ground

level, slope, the presence of fruiting trees and small water courses around camera-trap locations did not differ between FSC-certified and non-FSC concessions (Extended Data Fig. 3 and Extended Data Table 5). We also compared the presence and type of trails or paths around camera-trap locations, which did not differ significantly except for the number of elephant paths, which was higher in FSC-certified concessions (Extended Data Fig. 4 and Extended Data Table 5). As camera traps were installed randomly at the predetermined GPS locations on the nearest tree with 4 m visibility, finding a higher frequency of elephant paths in FSC-certified concessions was, in itself, an indication of higher elephant abundance in FSC-certified concessions. Potential seasonal influences are accounted for by the paired design. It is, however, important to note that encounter rates are a mixed measure of abundance and activity and we cannot disentangle whether changes in encounter rate are the result of changes in abundance, activity−movement per day−or both. Species' home ranges and movement patterns can change in response to disturbance, which can affect encounter rates. It is, however, unlikely that changes in activity solely make up the observed differences in encounter rates, given the consistency of the data in the three heaviest body mass classes. We also estimate relative biomass using encounter rates, which is a useful proxy to assess differences between forest management types but cannot be interpreted as true biomass (Extended Data Fig. 1).

## Conservation implications

Of central African tropical forest, 21% is designated for protection but only 15% of the species' ranges for central chimpanzees and the western lowland gorilla lie in protected areas[42,43]. More than half of these species' ranges and a large part of the ranges of other mammals, such as forest elephants, lie in logging concessions[26]. Protected areas are essential for conservation but sometimes lack the resources for effective control of illegal hunting[44,45]. Logging companies often do have the means to protect forests and have an economic incentive to do so. We did not compare mammal encounter rates in protected areas with the same metric in logging concessions ourselves. However, our observed encounter rates for large mammals, which are the first species to disappear as a result of hunting and poaching, in FSC-certified concessions were comparable to published data from recently monitored protected areas in the same region[46–48]. The ratio of large versus small forest antelopes in the FSC-certified concessions is furthermore comparable to such ratios in a protected area in the region with almost no hunting, whereas those in non-FSC concessions are far lower[49]. Although the first priority for species protection should be to maintain unlogged forests where there is effective law enforcement, our results challenge the notion that, at least for large-bodied mammals in WEA, logging is always disastrous for wildlife[50,51]. We show that, if selectively logged forests are properly managed, they can provide an important contribution to biodiversity: our results confirm that FSC-certified forests support far more larger and threatened species than do non-FSC forests. The results of this study are likely to be applicable to other logged tropical forests where hunting, through increased accessibility, poses a risk to forest mammals. This is because wildlife protection measures and law enforcement are applied across all FSC-certified forests, as part of the FSC principles, criteria and indicators for which FSC-certified companies are audited for compliance (Supplementary Tables 1 and 2). We infer this with caution as timber extraction volumes, concession size and shape, presence of public roads, population density and other characteristics may differ between concessions and thereby affect the impacts of FSC-certified forest management[52].

Most terrestrial protected areas are isolated[53] and increasing human modification and fragmentation of landscapes is limiting the ability of mammals to move[54]. Governments in forest-rich countries may enhance the effectiveness of conservation policies by requiring FSC certification in strategic locations, such as buffer zones around protected areas to reduce the edge-to-area ratio of the conservation landscape[55].

Non-FSC companies may also contribute to conservation, as they vary along a gradient of environmental and social responsibility[56]. This was, however, not the focus of our study. Concessions in our study region are large, often larger than 2,000 km², and together with protected areas they can substantially contribute to mammal conservation. Well-managed logging concessions can contribute to Sustainable Development Goal (SDG) 12 (sustainable consumption and production) and SDG 15 (life on land) by performing a strategic function in preserving habitats and landscape connectivity while allowing for responsible economic activity[57].

Our findings indicate that the requirements of FSC certification lead to effective mitigation of direct and indirect influences of logging on tropical forest mammals. The control of widespread and unsustainable hunting and poaching which is facilitated by the increased access to forests engendered by timber extraction is probably a key determinant of this impact. However, not all hunting is illegal and FSC certification protects customary rights to hunt non-protected species for subsistence. Sustainability of this practice is controlled by−among other requirements−controlling firearm permits, spatially assigning hunting zones and monitoring wildlife offtake. We believe that a strict set of requirements, control of compliance and regular enforcement, all integrally connected and ensured in the FSC system, are crucial for successful environmental protection through forest certification.

## The need to upscale certification

We present a clear, evidence-based message about the positive impact of FSC certification. We show that medium- to large-sized mammals−which play vital functions in forests−are more abundant in FSC-certified concessions than in non-FSC concessions. This study calls for action, reinforcing previous studies that called for more forest certification and land-use planning that takes conservation into account[14,26,43,58]. To protect large mammals, we urge that FSC certification or similar stringent schemes become the norm, as conventional logging is likely to result in half-empty forests dominated by rodents and other small species. To increase logging companies' interest in FSC certification, it is essential that sufficient demand is created for FSC-certified products by institutional and individual buyers. The information put forward by this study can play an important role in FSC's global strategy to leverage sustainable finance to reduce biodiversity loss, whereby certificate holders can be rewarded for the biodiversity benefits that they incur[59]. Rendering FSC-certified forests eligible for payments by biodiversity schemes, especially driven by government regulation[60], can contribute to fair valuation of standing forests. To ensure environmentally and socially responsible forest management practices[10], we strongly support the application of regulatory frameworks which stimulate and require the selling and buying of timber certified by FSC or similar stringent schemes.

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

## Methods

### Data collection

We set up arrays of camera traps from 2018 to 2021 in 14 logging concessions owned by 11 different companies (5 FSC and 6 non-FSC) in Gabon and the Republic of Congo (Fig. 1). Seven FSC-certified concessions were each paired to the closest non-FSC concession that was similar in terms of terrain and forest type[20]. All concessions are situated in a matrix of connected forests. In each pair of concessions, camera traps (Bushnell Trophy Cam HD for pairs 1–6 and Browning 2018 Spec Ops Advantage for pair 7) were deployed simultaneously to account for seasonal differences, for 2–3 months. There was one exception where Covid restrictions obliged the cameras to remain in place for longer (Extended Data Table 1). Camera-trap grid locations in each pair of concessions were chosen on the basis of similarity between potential drivers of mammal abundance, including distance to settlements, roads, rivers, protected areas, elevation (Extended Data Fig. 2 and Extended Data Table 5) and time since logging (2–10 years before our study), although some camera grids overlapped older logging blocks. Camera traps were set out in systematic, 1 km spaced grids with a random start point. On reaching the predetermined GPS locations, the first potential installation location was used where cameras had at least 4 m of visibility. This ensured that each grid was representative of environmental heterogeneity: that is, not specifically targeting or ignoring trails or other landscape elements that could influence detection[61]. The 1 km intercamera distance exceeds most species' home range sizes to avoid spatial autocorrelation. Species were not expected to migrate within the sampling duration of the study. Between 28 and 36 cameras were deployed in each concession, totalling 474 camera traps, distributed over 474 km$^2$ (Extended Data Table 1). Cameras were installed at a height of 30 cm to enable observations of mammals of all sizes. Cameras were programmed to take bursts of three photos to maximize the chance of detection and to take a photo every 12 h for correct calculation of active days in the event of a defect before the end of the deployment period. For each camera, we recorded whether there was an elephant path, skidder trail, small wildlife trail or an absence of a trail or path, in the field of view of each camera (Extended Data Fig. 4 and Extended Data Table 5). We also visually estimated forest visibility (0–10 m, 11–20 m, greater than 20 m), slope (0–5°, 5–20°, greater than 20°), presence of fruiting trees within 30 m and presence of small water courses within 50 m (Extended Data Fig. 3 and Extended Data Table 1). When approaching each predefined camera point, we counted cartridges, snares and hunting camps from 500 m before the camera up to its location. Various field teams were employed in different concessions and hence there may be some influence of interobserver bias of hunting observations between sites.

### Photo processing and data analysis

Camera-trap efforts yielded 1,278,853 photos, including 645,165 photos with animals. All photos were annotated in the program Wild.ID v.1.0.1. We identified animals up to the species level if photo quality permitted and otherwise designated the species as 'indet'[62]. As reliable species identification of small mammals is difficult, they were grouped into squirrels, rats and mice and shrews. Rare observations of humans, birds, bats, reptiles and domestic dogs were excluded from the analyses.

Observations of the same species that were at least 10 min apart were considered as separate detections. We assessed the sensitivity of this threshold by calculating the number of detections for intervals of 10, 30, 60 and 1,440 min, which all yielded proportionally similar numbers of observations across body mass classes (Supplementary Table 3). When several animals were observed, the number of individuals was determined by taking the highest number of individuals in a photo within the 10 min threshold. Sampling effort was defined as the number of camera days minus downtime due to malfunctioning cameras or obstruction of vision by vegetation.

Mammal behaviour may be different in hunted concessions, as mammals may be shyer of non-natural objects such as camera traps, which could in turn negatively affect their probability of detection. If this dynamic existed, shyness was assumed to fade over time with habituation to the materials, resulting in an increase of observations over time. We tested for an interaction between certification status and the number of observations over time using a linear model with a log-transformed number of observations for the first 68 days of all deployments, as that was the shortest concession deployment period, ensuring that all concessions were equally represented. We did not find that certification status was related to a trend in observations over time (Extended Data Fig. 5). We recognize, however, that other factors may have influenced detection probability, such as movement rates, which may be affected by hunting.

For each species for each concession, we calculated encounter rate, weighted by group size, as the number of observations divided by the sampling effort and we reported all findings using the metric 'observations per camera-trap day'. Encounter rate was calculated for all species combined, per body mass class, per IUCN Red List category[63] and per taxonomic group. Body mass of each species was determined by taking the mean across sexes[62]. Taxonomic groups Hyracoidea and Tubulidentata were excluded from the taxonomic analysis because of low sample sizes. Shrews were included as rodents in the taxonomy analysis even though they are formally not rodents because they are difficult to distinguish from mice. We consider this acceptable given that shrews are functionally very similar to rodents in the light of this study. To study the impact of certification on total estimated faunal biomass, the encounter rate of each species was multiplied by its average body mass divided by the sampling effort.

To assess whether encounter rates varied between FSC-certified and non-FSC concessions, we quantified the means of the paired concessions using linear mixed-effects models with concession pairs, concessions and cameras as random effects, whereby cameras were nested in concessions, in concession pairs, in a multilevel random effect structure. We allowed the means of concession pairs to vary between body mass class, IUCN Red List category and taxonomic group, if supported by model selection. We tested whether potential drivers of mammal abundance (Extended Data Figs. 2, 3 and 4) were important using a model-selection approach based on minimization of Bayesian information criterion values (Supplementary Table 4). We found that the inclusion of geographic covariates did not substantially improve the model for body mass classes, taxonomic groups and IUCN categories. Only for all mammals pooled together, the inclusion of elevation and distance to rivers resulted in slightly improved models but differences were negligible and did not support strong evidence for a significant influence of these covariates[64]. Quadratic geographic covariate terms and camera-trap site covariates did not result in better models. Pairwise comparisons were multivariate $t$ adjusted. We used two-sided Wilcoxon signed-rank tests for all other analyses (Extended Data Table 5). Statistical analyses were performed in R v.4.2.2.

### Reporting summary

Further information on research design is available in the Nature Portfolio Reporting Summary linked to this article.

## Data availability

The data that support the findings of this study are available in the Zenodo repository under https://doi.org/10.5281/zenodo.10061155 (ref. 65). Source data are provided with this paper.

## Code availability

R code for statistical analyses and data tables are available in the Zenodo repository under https://doi.org/10.5281/zenodo.10061155 (ref. 65).

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

**Acknowledgements** We thank the logging companies for access to their concessions, 263 people from WEA for fieldwork assistance and 23 students and assistants for data processing. We also thank Y. Hautier for his insights concerning the statistical analyses. The work was carried out with permission from the Gabonese Centre National de la Recherche Scientifique et Technologique (CENAREST) under research permit no. AV AR0046/18 and the Congolese Institut National de Recherche Forestière under research permit nos. 219 and 126 issued by the National Forest Research Institute (IRF) of Congo with the help of the Wildlife Conservation Society (WCS) Congo, under no. 219MRSIT/IRF/DG/DS on 17 July 2019 and the extension under no. 126MRSIT/IRF/DG/DS on 4 August 2020. J.A.Z. received support for this work from the Dutch Research Council NWO through the graduate programme Nature Conservation, Management and Restoration (grant no. 022.006.011), Programme de Promotion de l'Exploitation Certifiée des Forêts (PPECF) de la COMIFAC (à travers la KfW) under grant no. C146, WWF Netherlands, WWF Germany and the Prince Bernhard Chair for International Nature Conservation of Utrecht University.

**Author contributions** J.A.Z., M.v.K., J.v.d.W. and G.B.T. conceptualized this article. J.A.Z., E.A.M.G. and H.F.D.Z. were responsible for data curation. J.A.Z. and E.A.M.G. conducted the formal analysis. J.A.Z., M.v.K., E.A.M.G., E.H.M.S. and P.A.V. developed the methodology. J.A.Z. and H.F.D.Z. undertook investigations. J.A.Z. and E.A.M.G. created the visualizations. P.A.V., J.A.Z., M.v.K. and J.v.d.W. acquired funding. J.A.Z. and G.B.T. were responsible for project administration. J.A.Z., M.v.K., G.B.T., E.H.M.S., P.A.V. and F.M. supervised the work. J.A.Z. wrote the original draft manuscript. J.A.Z., M.v.K., J.v.d.W., F.M., G.B.T., P.A.V. and E.H.M.S. reviewed and edited the final article.

**Competing interests** J.A.Z. is an unpaid individual member of the FSC Environmental chamber, sub-chamber North. G.B.T. is an unpaid individual member of the FSC Environmental chamber, sub-chamber South and, since 2018, also a member of an advisory committee to the Board of Directors of FSC, the Policy and Standard Committee. J.v.d.W. and H.F.D.Z. have unpaid institutional membership of FSC through WWF International. The remaining authors declare no competing interests.

**Additional information**
**Correspondence and requests for materials** should be addressed to Joeri A. Zwerts.

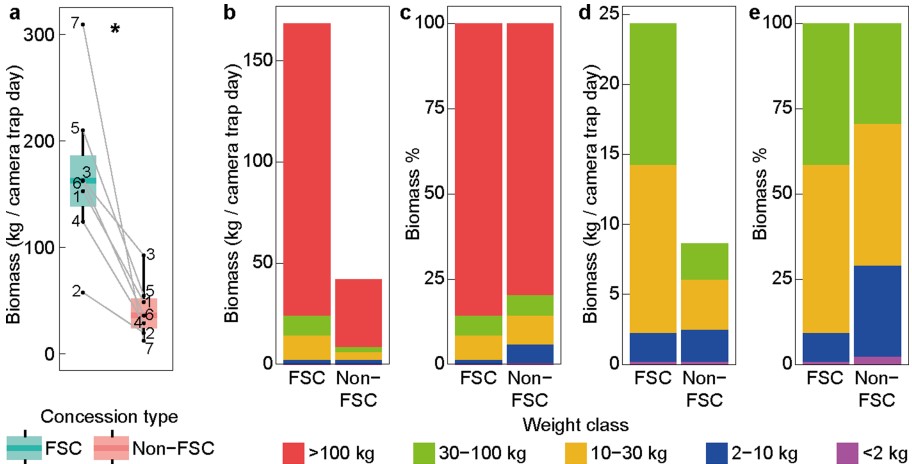

**Extended Data Fig. 1 | Estimated faunal biomass derived from mammal encounter rates.** (**a**) Estimated faunal biomass was higher (p = 0.016) in FSC-certified (n = 7) than in non-FSC concessions (n = 7). Numbers represent paired FSC-certified and non-FSC concessions linked by grey lines. Data is represented as a boxplot, where central lines represent medians and lower and upper lines correspond to the first and third quartiles, whiskers reflect 1.5 times the interquartile range. Two-sided Wilcoxon signed-rank, *: p < 0.05. Panels (b–e) represent the contributions of different body mass classes to the estimated faunal biomass derived from mammal encounter rates in FSC-certified (n = 7) and non-FSC concessions (n = 7). (**b**) in kg / camera-trap day; (**c**) as a proportion of total faunal biomass; (**d**) in kg/day for species up to 100 kg; (**e**) as a proportion of the total faunal biomass for species up to 100 kg. FSC-certified concessions had higher overall biomass whereby mammals weighing more than 10 kg made up a larger proportion of the total biomass than in non-FSC concessions.

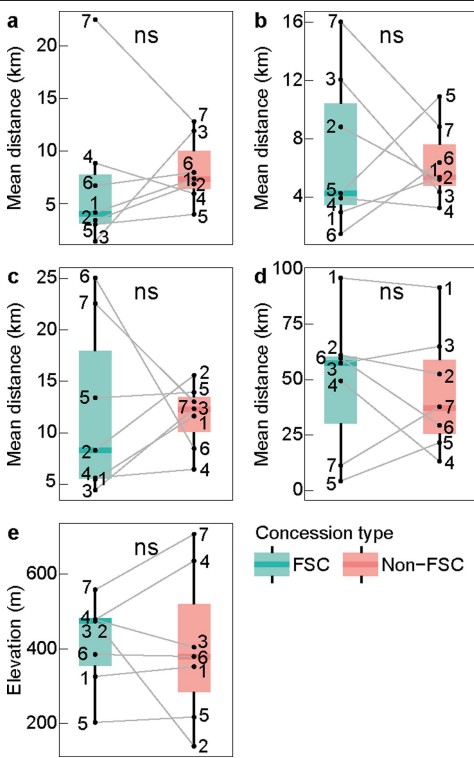

**Extended Data Fig. 2 | Geographic covariates.** (**a**) Distance to roads, (**b**) rivers, (**c**) human settlements, (**d**) and protected areas, as well as (**e**) elevation, did not differ significantly between camera locations in FSC-certified (n = 7) and non-FSC concessions (n = 7). Numbers represent paired FSC-certified and non-FSC concessions linked by grey lines. Data are represented as boxplots, where central lines represent medians and lower and upper lines correspond to the first and third quartiles, whiskers reflect 1.5 times the interquartile range. Two-sided Wilcoxon signed-rank, ns: p > 0.05. Exact p-values are summarized in Extended Data Table 4.

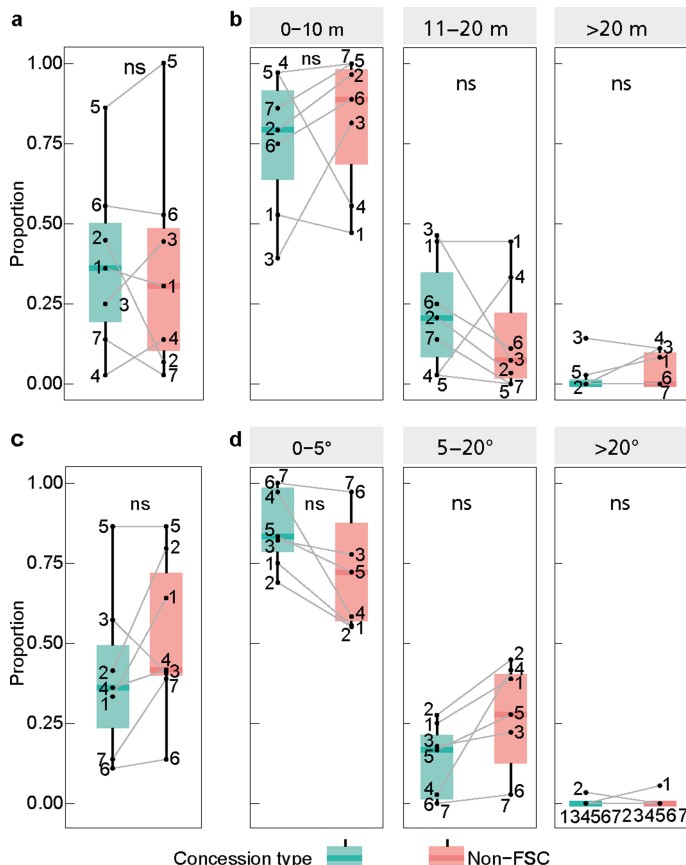

**Extended Data Fig. 3 | Camera trap site covariates.** (**a**) The presence of
fruiting trees within 30 m, (**b**) visibility, (**c**) the presence of small water courses
within 50 m distance and (**d**) slope, expressed in proportions, did not differ
significantly between camera locations in FSC-certified (n = 7) and non-FSC
concessions (n = 7). Numbers represent paired FSC-certified and non-FSC
concessions linked by grey lines. Data are represented as boxplots, where
central lines represent medians and lower and upper lines correspond to the
first and third quartiles, whiskers reflect 1.5 times the interquartile range.
Two-sided Wilcoxon signed-rank, ns: p > 0.05. Exact p-values are summarized
in Extended Data Table 4.

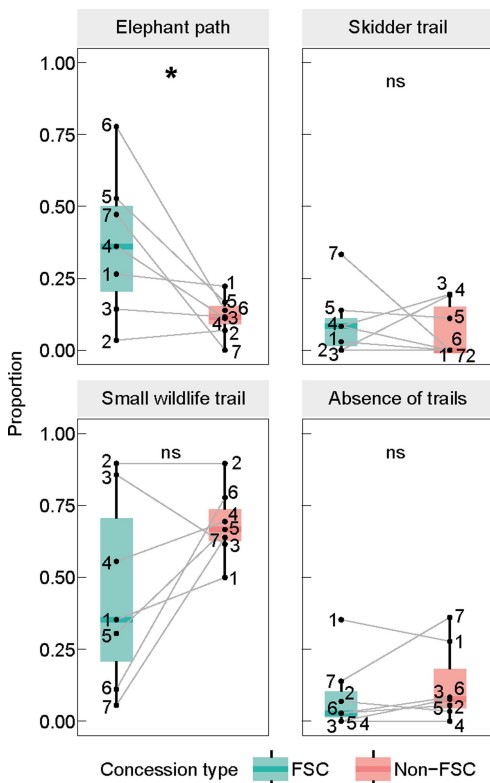

**Extended Data Fig. 4 | The presence of trails or paths in the field of view of randomly placed cameras.** Each camera trap installation location was characterized as either an elephant path, skidder trail, small wildlife trail or as an absence of a trail or path. Only elephant paths, expressed in proportions, were encountered more often in FSC-certified concessions (n = 7) than in non-FSC concessions (n = 7), whereas the presence or absence of the other three types of installation locations was equivalent between the two forest management types. Camera trap sites were selected as the closest location from the predetermined GPS locations with both a suitable tree and a minimum of four metres visibility. Following this method, randomly encountering more elephant paths is in itself an indication of higher elephant abundances in FSC-certified concessions. Numbers represent paired FSC-certified and non-FSC concessions linked by grey lines. Data are represented as boxplots, where central lines represent medians and lower and upper lines correspond to the first and third quartiles, whiskers reflect 1.5 times the interquartile range. Two-sided Wilcoxon signed-rank, *: p <= 0.05, ns: p > 0.05. Exact p-values are summarized in Extended Data Table 4.

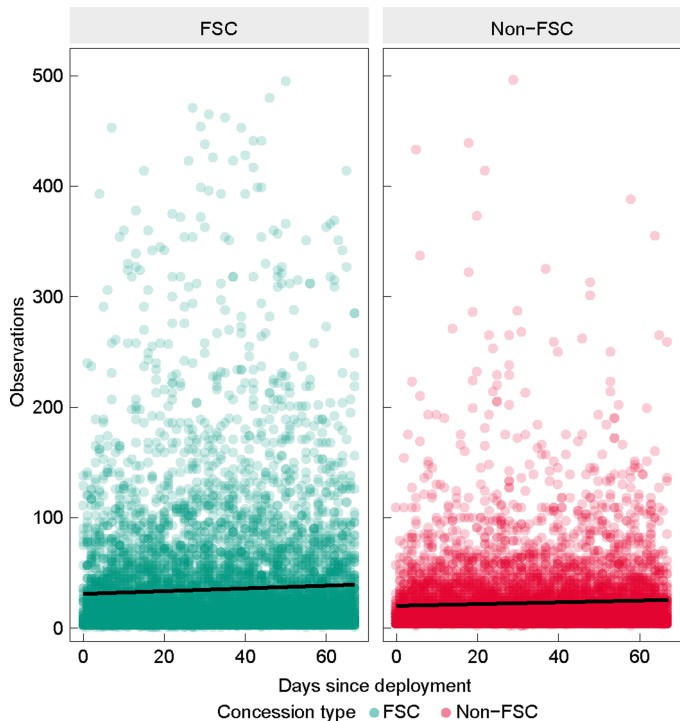

**Extended Data Fig. 5 | Observations over time.** This analysis explored whether variation in hunting induced mammal shyness for non-natural objects influenced detection differentially in FSC-certified (n = 7) and non-FSC concessions (n = 7). We did not find support for an effect of certification status on the number of observations over time. Linear model: p = 0.892.

**Extended Data Table 1 | Camera trap deployment sites and periods**

| Pair | Country | Type | Number of cameras | Deployment period | Total deployment time (days) | Effort (days) |
|---|---|---|---|---|---|---|
| 1 | Gabon | FSC | 36 | Dec 2018 – Mar 2019 | 2,597 | 2,453 |
| 1 | Gabon | Non-FSC | 36 | Nov 2018 – Mar 2019 | 1,960 | 1,960 |
| 2 | Gabon | Non-FSC | 29 | Apr 2019 – Jul 2019 | 2,128 | 2,128 |
| 2 | Gabon | FSC | 29 | Mar 2019 – Jun 2019 | 2,070 | 2,030 |
| 3 | Gabon | Non-FSC | 28 | Apr 2019 – Jul 2019 | 1,264 | 1,258 |
| 3 | Gabon | FSC | 28 | Mar 2019 - Jun 2019 | 1,172 | 1,087 |
| 4 | Gabon | FSC | 36 | Jul 2019 – Oct 2019 | 3,041 | 3,007 |
| 4 | Gabon | Non-FSC | 36 | Jun 2019 - Oct 2019 | 3,071 | 2,962 |
| 5 | Gabon | FSC | 36 | Oct 2019 – Jan 2020 | 2,186 | 2,148 |
| 5 | Gabon | Non-FSC | 36 | Nov 2019 – Jan 2020 | 2,092 | 2,087 |
| 6 | Congo | FSC | 36 | Mar 2020 – Oct 2020 | 5,277 | 4,554 |
| 6 | Congo | Non-FSC | 36 | Feb 2020 – Oct 2020 | 5,558 | 5,208 |
| 7 | Congo | FSC | 36 | Mar 2021 – Jun 2021 | 2,537 | 2,532 |
| 7 | Congo | Non-FSC | 36 | Mar 2021 – Jun 2021 | 2,132 | 2,132 |
| **Total** | | | **474** | | **37,085** | **35,546** |
| **Average per site** | | | **34** | | **2,649** | **2,539** |

The number of cameras was defined as the number of cameras that were deployed in a concession. The period of deployment was noted as the month and year the first camera trap was placed and last camera trap was recuperated. The total deployment time was calculated by taking the sum of all active camera-trap days per site. The effort is the total deployment time of a site minus the time camera traps were malfunctioning or covered by vegetation. All sites were deployed for two to three months with one exception where Covid travel restrictions resulted in the cameras remaining in place for longer. The companies are not named to assure anonymity: this was a prerequisite for several companies to participate in the study.

## Extended Data Table 2 | Observed mammals

| Species names | Taxonomy | Body mass (kg) | IUCN | Observed mean encounter rate FSC (± SEM) | Observed mean encounter rate non-FSC (± SEM) |
|---|---|---|---|---|---|
| *Cephalophus callipygus* | Even-toed ungulates | 21.35 | LC | **0.322 (± 0.138)** | 0.065 (± 0.026) |
| *Philantomba monticola* | Even-toed ungulates | 6.25 | LC | 0.229 (± 0.036) | **0.285 (± 0.048)** |
| Squirrels | Rodents | 0.27 | LC | **0.186 (± 0.022)** | 0.158 (± 0.033) |
| *Atherurus africanus* | Rodents | 3.25 | LC | 0.105 (± 0.027) | **0.117 (± 0.012)** |
| Mice and shrews | Rodents | 0.010 | LC | 0.083 (± 0.027) | **0.097 (± 0.037)** |
| *Potamochoerus porcus* | Even-toed ungulates | 80 | LC | **0.082 (± 0.044)** | 0.018 (± 0.005) |
| *Cephalophus dorsalis* | Even-toed ungulates | 19.75 | NT | **0.077 (± 0.032)** | 0.034 (± 0.009) |
| *Cephalophus ogilbyi* | Even-toed ungulates | 17 | LC | **0.075 (± 0.053)** | 0.023 (± 0.015) |
| *Cricetomys emini* | Rodents | 1.2 | LC | 0.067 (± 0.033) | **0.092 (± 0.031)** |
| *Loxodonta cyclotis* | Elephants | 2150 | CR | **0.066 (± 0.012)** | 0.015 (± 0.005) |
| *Cephalophus* indet | Even-toed ungulates | 18.32 | LC | **0.062 (± 0.018)** | 0.033 (± 0.013) |
| *Cephalophus silvicultor* | Even-toed ungulates | 62.5 | NT | **0.041 (± 0.014)** | 0.008 (± 9.44e-04) |
| *Mandrillus sphinx* | Primates | 18.5 | VU | **0.031 (± 0.015)** | 0.019 (± 0.010) |
| Rats | Rodents | 0.14 | LC | 0.027 (± 0.017) | **0.060 (± 0.029)** |
| *Genetta servalina* | Carnivores | 1.85 | LC | **0.026 (± 0.006)** | 0.021 (± 0.005) |
| *Bdeogale nigripes* | Carnivores | 2.75 | LC | **0.021 (± 0.005)** | 0.009 (± 0.003) |
| *Hyemoschus aquaticus* | Even-toed ungulates | 10.85 | LC | **0.021 (± 0.008)** | 0.011 (± 0.006) |
| *Pan troglodytes* | Primates | 31.5 | EN | **0.019 (± 0.006)** | 0.011 (± 0.003) |
| *Xenogale naso* | Carnivores | 3.6 | LC | **0.019 (± 0.013)** | 0.007 (± 0.002) |
| *Atilax paludinosus* | Carnivores | 3.5 | LC | **0.014 (± 0.006)** | 0.006 (± 0.002) |
| *Cephalophus leucogaster* | Even-toed ungulates | 17.5 | NT | **0.012 (± 0.003)** | 0.002 (± 8.27e-04) |
| *Gorilla gorilla* | Primates | 116.5 | CR | **0.010 (± 0.003)** | 0.004 (± 0.001) |
| *Cephalophus nigrifrons* | Even-toed ungulates | 16 | LC | **0.008 (± 0.005)** | 4.02e-04 (± 4.02e-04) |
| *Cercocebus agilis* | Primates | 7.83 | LC | **0.008 (± 0.005)** | 0.002 (± 0.002) |
| *Caracal aurata* | Carnivores | 10.1 | VU | **0.006 (± 0.002)** | 6.84e-04 (± 2.12e-04) |
| *Nandinia binotata* | Carnivores | 2.6 | LC | 0.005 (± 0.001) | **0.007 (± 0.002)** |
| *Panthera pardus* | Carnivores | 53.25 | VU | **0.004 (± 0.001)** | 0.002 (± 6.14e-04) |
| *Phataginus tricuspis* | Pangolins | 2.3 | EN | 0.004 (± 0.002) | **0.004 (± 0.002)** |
| *Crossarchus platycephalus* | Carnivores | 1.25 | LC | 0.004 (± 0.002) | **0.006 (± 0.002)** |
| *Cercocebus torquatus* | Primates | 8.13 | EN | **0.003 (± 0.003)** | 0.002 (± 0.002) |
| *Phataginus tetradactyla* | Pangolins | 2.9 | VU | 0.002 (± 8.49e-04) | **0.004 (± 0.001)** |
| *Smutsia gigantea* | Pangolins | 32.5 | EN | **0.002 (± 4.07e-04)** | 0.001 (± 4.12e-04) |
| *Cercopithecus nictitans* | Primates | 5.53 | NT | **0.001 (± 7.86e-04)** | 3.62e-04 (± 3.32e-04) |
| *Syncerus caffer* | Even-toed ungulates | 637.5 | NT | **8.52e-04 (± 3.98e-04)** | 7.05e-04 (± 3.32e-04) |
| *Civettictis civetta* | Carnivores | 13.5 | LC | **6.38e-04 (± 5.10e-04)** | 0 |
| *Orycteropus afer* | Tubulidentata* | 61 | LC | **5.55e-04 (± 5.20e-04)** | 0 |
| *Cercopithecus* indet | Primates | 4.13 | NT | **4.43e-04 (± 3.90e-04)** | 1.91e-04 (± 1.04e-04) |
| *Euoticus elegantulus* | Primates | 0.32 | LC | 4.39e-04 (± 2.11e-04) | **4.59e-04 (± 4.05e-04)** |
| *Mellivora capensis* | Carnivores | 9.85 | LC | **3.97e-04 (± 2.73e-04)** | 0 |
| *Lophocebus albigena* | Primates | 7 | VU | **2.26e-04 (± 2.26e-04)** | 0 |
| *Poiana richardsonii* | Carnivores | 0.60 | LC | **1.88e-04 (± 1.41e-04)** | 0 |
| *Tragelaphus spekii* | Even-toed ungulates | 69 | LC | 1.57e-04 (± 1.57e-04) | **0.001 (± 0.001)** |
| *Tragelaphus eurycerus* | Even-toed ungulates | 275.75 | NT | **1.25e-04 (± 1.25e-04)** | 0 |
| *Dendrohyrax dorsalis* | Hyracoidea* | 3.15 | LC | **1.18e-04 (± 7.81e-05)** | 0 |
| *Cercopithecus cephus* | Primates | 3.35 | LC | 8.96e-05 (± 6.14e-05) | **7.81e-04 (± 2.91e-04)** |
| *Arctocebus aureus* | Primates | 0.235 | LC | 7.89e-05 (± 5.24e-05) | **9.65e-05 (± 9.65e-05)** |
| *Aonyx congicus* | Carnivores | 20 | NT | **5.64e-05 (± 5.64e-05)** | 0 |
| *Thryonomys swinderianus* | Rodents | 4.35 | LC | 3.14e-05 (± 3.14e-05) | **5.49e-05 (± 5.49e-05)** |
| *Hylochoerus meinertzhageni* | Even-toed ungulates | 178.75 | LC | 0 | **2.54e-04 (± 2.24e-04)** |
| *Galagoides thomasi* | Primates | 0.10 | LC | 0 | **1.15e-04 (± 7.59e-05)** |
| *Perodicticus potto* | Primates | 1.2 | NT | 0 | **8.23e-05 (± 8.23e-05)** |
| *Sciurocheirus gabonensis* | Primates | 0.26 | LC | 0 | **4.82e-05 (± 4.82e-05)** |
| *Cercopithecus pogonias* | Primates | 3.53 | NT | 0 | **2.74e-05 (± 2.74e-05)** |
| *Colobus guereza* | Primates | 9.28 | LC | 0 | **2.74e-05 (± 2.74e-05)** |
| *Colobus satanas* | Primates | 9.9 | VU | 0 | **2.74e-05 (± 2.74e-05)** |

Encounter rates (observations per camera-trap day) of observed mammals in FSC-certified (n=7) and non-FSC concessions (n=7), ranked in descending order of encounter rate in FSC-certified concessions. Per certification type the highest encounter rate of a species is depicted in bold. Body mass (categorized in five classes: <2 kg, 2–10 kg, 10–30 kg, 30–100 kg, >100 kg) was retrieved from Kingdon (2015). IUCN Red List category was retrieved from the IUCN Red List[63]. IUCN abbreviations: CR = Critically Endangered, EN = Endangered, VU = Vulnerable, NT = Near Threatened, LC = Least Concern. * = Have less than 10 observations and are therefore not included as separate taxonomic groups in the taxonomy analysis.

**Extended Data Table 3 | Linear Mixed Model Type III analysis of variance tables with Satterthwaite's method**

|  | Fixed effects | Mean Square | Numerator Degrees of freedom | Denominator Degrees of Freedom | F value | P value (>F) |
|---|---|---|---|---|---|---|
| Total encounter rate | FSC certification | 0.729 | 1 | 12.049 | 6.7787 | 0.02301 * |
| Body mass classes | FSC certification | 37.200 | 1 | 11.12 | 35.796 | 8.758e-05 *** |
|  | Body mass class | 82.112 | 4 | 6.20 | 79.014 | 1.938e-05 *** |
|  | FSC certification: Body mass class | 46.339 | 4 | 1754.5 | 44.590 | < 2.2e-16 *** |
| IUCN Red List categories | FSC certification | 28.776 | 1 | 11.39 | 39.100 | 5.349e-05 *** |
|  | IUCN Red List category | 200.304 | 4 | 6.08 | 272.169 | 5.609e-07 *** |
|  | FSC certification: IUCN Red List category | 14.541 | 4 | 1753.46 | 19.759 | 6.344e-16 *** |
| Taxonomic groups | FSC certification | 13.720 | 1 | 11.39 | 15.646 | 0.002107 ** |
|  | Taxonomic group | 419.550 | 5 | 7.02 | 478.351 | 1.002e-08 *** |
|  | FSC certification: Taxonomic group | 23.420 | 5 | 2197.75 | 26.699 | 2.2e-16 *** |

Each model included the concessions, concession pairs and cameras as random effects. ***: p<0.001, **: p<0.01, * p<0.05.

**Extended Data Table 4 | Pairwise comparisons and descriptive statistics of mammal encounter rates in paired FSC-certified (n=7) and non-FSC concessions (n=7)**

| Grouping variable and statistical test | Grouping classes (Numbers between brackets indicate the number of species per class) | Test statistic and multivariate-t adjusted p-values | FSC estimated mean encounter rate and confidence limits | Non-FSC estimated mean encounter rate and confidence limits |
|---|---|---|---|---|
| Body mass, pairwise comparisons | >100 kg (5) *** | p < 0.001 | 0.046 (0.035-0.060) | 0.017 (0.013-0.022) |
| | 30-100 kg (7) *** | p < 0.001 | 0.070 (0.050-0.100) | 0.028 (0.020-0.039) |
| | 10-30 kg (11) *** | p < 0.001 | 0.338 (0.200-0.571) | 0.096 (0.057-0.163) |
| | 2-10 kg (20) | p = 0.743 | 0.235 (0.160-0.345) | 0.247 (0.169-0.362) |
| | <2 kg (13) | p = 0.995 | 0.215 (0.150-0.318) | 0.247 (0.173-0.355) |
| IUCN Red List categories, pairwise comparisons | Critically Endangered (CR) (2) *** | p < 0.001 | 0.045 (0.034-0.059) | 0.017 (0.013-0.022) |
| | Endangered (EN) (4) | p = 0.194 | 0.024 (0.019-0.030) | 0.019 (0.015-0.024) |
| | Vulnerable (VU) (6) | p = 0.178 | 0.026 (0.016-0.041) | 0.020 (0.013-0.032) |
| | Near Threatened (NT) (10) *** | p < 0.001 | 0.071 (0.043-0.116) | 0.031 (0.019-0.050) |
| | Least Concern (LC) (34) * | p = 0.036 | 0.993 (0.077-1.28) | 0.712 (0.552-0.918) |
| Taxonomy, pairwise comparisons | Elephants (1) *** | p < 0.001 | 0.038 (0.030-0.047) | 0.015 (0.012-0.018) |
| | Primates (17) ** | p = 0.002 | 0.048 (0.035-0.065) | 0.027 (0.020-0.037) |
| | Even-toed Ungulates (14) *** | p < 0.001 | 0.564 (0.410-0.776) | 0.278 (0.202-0.383) |
| | Carnivores (12) * | p = 0.027 | 0.067 (0.046-0.098) | 0.045 (0.031-0.066) |
| | Pangolins (3) | p = 0.908 | 0.014 (0.011-0.018) | 0.016 (0.012-0.020) |
| | Rodents (6) | p = 0.336 | 0.244 (0.184-0.325) | 0.305 (0.229-0.407) |

Pairwise comparisons were multivariate t adjusted. ***: p<0.001, **: p<0.01, * p<0.05.

**Extended Data Table 5 | Hunting signs, geographic covariates and camera trap site covariates in FSC-certified (n=7) and non-FSC concessions (n=7)**

| Covariate | Categories | Test statistic and uncorrected p-values | FSC medians and interquartile range | Non-FSC medians and interquartile range |
|---|---|---|---|---|
| Hunting signs (in proportions) | - | **V = 26, p = 0.036** | 0 (0.045) | 0.139 (0.245) |
| Distance to roads (m) | - | V = 9, p = 0.469 | 4063 (4587) | 7336 (3558) |
| Distance to rivers (m) | - | V = 17, p = 0.688 | 4259 (6998) | 5355 (2813) |
| Distance to human settlements (m) | - | V = 13, p = 0.938 | 8304 (12436) | 12334 (3395) |
| Distance to protected areas (m) | - | V = 17, p = 0.688 | 57372 (30014) | 37341 (33290) |
| Elevation (m) | - | V = 12, p = 0.81 | 474 (122) | 380 (235) |
| The presence of trails or paths (in proportions) | Elephant path | **V = 28, p = 0.016** | 0.361 (0.296) | 0.115 (0.063) |
| | Skidder trail | V = 12, p = 0.833 | 0.083 (0.096) | 0 (0.152) |
| | Small wildlife trail | V = 4, p = 0.109 | 0.353 (0.498) | 0.667 (0.109) |
| | Absence of trails | V = 8, p = 0.675 | 0.028 (0.090) | 0.077 (0.136) |
| Forest visibility (in proportions) | 0-10 m | V = 9, p = 0.469 | 0.793 (0.278) | 0.889 (0.298) |
| | 11-20 m | V = 20, p = 0.375 | 0.207 (0.264) | 0.074 (0.205) |
| | >20m | V = 1, p = 0.414 | 0 (0.014) | 0 (0.097) |
| Camera trap site slope (in proportions) | 0-5° | V = 15, p = 0.402 | 0.833 (0.2) | 0.722 (0.306) |
| | 5-20° | V = 4, p = 0.208 | 0.167 (0.2) | 0.278 (0.278) |
| | >20° | V = 1, p = 1 | 0 | 0 |
| Presence of fruiting trees within 30 m (in proportions) | - | V = 13.5, p = 1 | 0.361 (0.307) | 0306 (0.382) |
| Presence of small water courses within 50 m (in proportions) | - | V = 3, p = 0.142 | 0.361 (0.256) | 0.417 (0.318) |

Two-sided Wilcoxon signed-rank, Bold: p<0.05, Underscore: p<0.1.

# Reporting Summary

## Statistics

For all statistical analyses, confirm that the following items are present in the figure legend, table legend, main text, or Methods section.

| n/a | Confirmed | |
|---|---|---|
| ☐ | ☒ | The exact sample size (*n*) for each experimental group/condition, given as a discrete number and unit of measurement |
| ☐ | ☒ | A statement on whether measurements were taken from distinct samples or whether the same sample was measured repeatedly |
| ☐ | ☒ | The statistical test(s) used AND whether they are one- or two-sided *Only common tests should be described solely by name; describe more complex techniques in the Methods section.* |
| ☐ | ☒ | A description of all covariates tested |
| ☐ | ☒ | A description of any assumptions or corrections, such as tests of normality and adjustment for multiple comparisons |
| ☐ | ☒ | A full description of the statistical parameters including central tendency (e.g. means) or other basic estimates (e.g. regression coefficient) AND variation (e.g. standard deviation) or associated estimates of uncertainty (e.g. confidence intervals) |
| ☐ | ☒ | For null hypothesis testing, the test statistic (e.g. *F*, *t*, *r*) with confidence intervals, effect sizes, degrees of freedom and *P* value noted *Give P values as exact values whenever suitable.* |
| ☒ | ☐ | For Bayesian analysis, information on the choice of priors and Markov chain Monte Carlo settings |
| ☐ | ☒ | For hierarchical and complex designs, identification of the appropriate level for tests and full reporting of outcomes |
| ☒ | ☐ | Estimates of effect sizes (e.g. Cohen's *d*, Pearson's *r*), indicating how they were calculated |

*Our web collection on statistics for biologists contains articles on many of the points above.*

## Software and code

Policy information about availability of computer code

| Data collection | No software was used for data collection, all photos were annotated in the program Wild.ID. |
|---|---|
| Data analysis | Statistical analyses were performed in R version 4.2.2. R code for statistical analyses and data tables is available in the Zenodo repository under https://doi.org/10.5281/zenodo.10061155. |

For manuscripts utilizing custom algorithms or software that are central to the research but not yet described in published literature, software must be made available to editors and reviewers. We strongly encourage code deposition in a community repository (e.g. GitHub). See the Nature Portfolio guidelines for submitting code & software for further information.

## Data

Policy information about availability of data

All manuscripts must include a data availability statement. This statement should provide the following information, where applicable:
- Accession codes, unique identifiers, or web links for publicly available datasets
- A description of any restrictions on data availability
- For clinical datasets or third party data, please ensure that the statement adheres to our policy

The data that support the findings of this study is available in the Zenodo repository under https://doi.org/10.5281/zenodo.10061155

# Research involving human participants, their data, or biological material

Policy information about studies with human participants or human data. See also policy information about sex, gender (identity/presentation), and sexual orientation and race, ethnicity and racism.

| | |
|---|---|
| Reporting on sex and gender | N/A |
| Reporting on race, ethnicity, or other socially relevant groupings | N/A |
| Population characteristics | N/A |
| Recruitment | N/A |
| Ethics oversight | N/A |

Note that full information on the approval of the study protocol must also be provided in the manuscript.

# Field-specific reporting

Please select the one below that is the best fit for your research. If you are not sure, read the appropriate sections before making your selection.

☐ Life sciences  ☐ Behavioural & social sciences  ☒ Ecological, evolutionary & environmental sciences

For a reference copy of the document with all sections, see nature.com/documents/nr-reporting-summary-flat.pdf

# Ecological, evolutionary & environmental sciences study design

All studies must disclose on these points even when the disclosure is negative.

| | |
|---|---|
| Study description | We used camera traps to assess whether FSC certification can mitigate the negative effects of timber extraction on wildlife by studying the encounter rate of a broad range of mammal species across multiple sites. |
| Research sample | We compared small to large-sized mammal observations across seven paired FSC-certified and non-FSC concessions in Gabon and the Republic of Congo. Gabon and the Republic of Congo lie within Western Equatorial Africa (WEA), and we included all companies that were FSC-certified between 2018 and 2021 in this region, with the exception of one which refused to allow access. WEA is particularly suitable for these analyses, as its forests are reasonably intact and logging concessions are embedded in a matrix of contiguous forest, which are therefore mostly devoid of influences other than the effects of logging and hunting. Wild meat hunting is pervasive in WEA. Logging increases hunting pressure by increased access (logging roads) and by the arrival of people working in the concessions in once-remote forests. By ensuring close spatial pairing of the FSC-certified and non-FSC concessions we reduced the influence of regional landscape heterogeneity. |
| Sampling strategy | We set up arrays of camera traps from 2018 to 2021 in 14 logging concessions owned by 11 different companies (5 FSC and 6 non-FSC) in Gabon and the Republic of Congo. We included all companies that were FSC-certified between 2018 and 2021 in this region, with the exception of one which refused to allow access. Seven FSC-certified concessions were each paired to the closest non-FSC concession that was similar in terms of terrain and forest type. All concessions are situated in a matrix of connected forests. Within each pair of concessions, camera traps (Bushnell Trophy Cam HD for pairs 1 - 6 and Browning 2018 Spec Ops Advantage for pair 7) were deployed simultaneously to account for seasonal differences, for two to three months. There was one exception where Covid restrictions obliged the cameras to remain in place for longer. Camera trap grid locations within each pair of concessions were chosen based on similarity between potential drivers of mammal abundance, including distance to settlements, roads, rivers, protected areas, elevation and time since logging (2-10 years before our study), although some camera grids overlapped older logging blocks. |
| Data collection | Camera traps were set out in systematic, one-kilometre spaced grids with a random start point. Upon reaching the predetermined GPS locations, the first potential installation location was used where cameras had at least 4 metres of visibility. This ensured that each grid was representative of environmental heterogeneity: i.e. not specifically targeting nor ignoring trails or other landscape elements that could influence detection. The one-kilometre inter-camera distance exceeds most species' home range sizes to avoid spatial autocorrelation. Neither were species expected to migrate within the sampling duration of the study. Between 28 to 36 cameras were deployed in each concession, totalling up to 474 camera traps, distributed over 474 km2. Cameras were installed at a height of 30 cm to enable observations of mammals of all sizes while ensuring that each camera had at least four metres visibility in front of it. Cameras were programmed to take bursts of three photos to maximize the chance of detection and to take a photo every 12 hours for correct calculation of active days in the event of a defect before the end of the deployment period. For each camera, we recorded whether there was an elephant path skidder trail, small wildlife trail, or none of the above within each camera's field of view. We also visually estimated forest visibility (0-10m / 11-20m / >20m), slope (0-5° / 5-20° / >20°), presence of fruiting trees within 30 m and presence of small water courses within 50 m. When approaching each predefined camera point, we counted cartridges, snares and hunting camps from 500 m before the camera up to its location. Various field teams were employed in different sites and hence there may be some influence interobserver bias of hunting observations between sites. |
| Timing and spatial scale | We set up arrays of camera traps from 2018 to 2021. Camera traps were deployed simultaneously to account for seasonal |

| Timing and spatial scale | differences, for two to three months. There was one exception where Covid restrictions obliged the cameras to remain in place for longer. |
|---|---|
| Data exclusions | No data was excluded from the analysis. |
| Reproducibility | All camera trap data is available and can be reanalyzed to ensure the reproducibility. |
| Randomization | Camera trap grid locations within each pair of concessions were chosen based on similarity between potential drivers of mammal abundance, including distance to settlements, roads, rivers, protected areas, elevation and time since logging (2-10 years before our study), although some camera grids overlapped older logging blocks. Camera traps were set out in systematic, one-kilometre spaced grids with a random start point. Upon reaching the predetermined GPS locations, the first potential installation location was used where cameras had at least 4 metres of visibility. This ensured that each grid was representative of environmental heterogeneity: i.e. not specifically targeting nor ignoring trails or other landscape elements that could influence detection. The one-kilometre inter-camera distance exceeds most species' home range sizes to avoid spatial autocorrelation. Neither were species expected to migrate within the sampling duration of the study. Between 28 to 36 cameras were deployed in each concession, totalling up to 474 camera traps, distributed over 474 km2. Cameras were installed at a height of 30 cm to enable observations of mammals of all sizes while ensuring that each camera had at least four metres visibility in front of it. Cameras were programmed to take bursts of three photos to maximize the chance of detection and to take a photo every 12 hours for correct calculation of active days in the event of a defect before the end of the deployment period. For each camera, we recorded whether there was an elephant path skidder trail, small wildlife trail, or none of the above within each camera's field of view. We also visually estimated forest visibility (0-10m / 11-20m / >20m), slope (0-5° / 5-20° / >20°), presence of fruiting trees within 30 m and presence of small water courses within 50 m. When approaching each predefined camera point, we counted cartridges, snares and hunting camps from 500 m before the camera up to its location. Various field teams were employed in different sites and hence there may be some influence interobserver bias of hunting observations between sites. |
| Blinding | Blinding was not relevant for our study as all mammal observations are linked to locations. |

Did the study involve field work? ☒ Yes ☐ No

# Field work, collection and transport

| Field conditions | Within each pair of concessions, camera traps (Bushnell Trophy Cam HD for pairs 1 - 6 and Browning 2018 Spec Ops Advantage for pair 7) were deployed simultaneously to account for seasonal differences, for two to three months. There was one exception where Covid restrictions obliged the cameras to remain in place for longer. Camera trap grid locations within each pair of concessions were chosen based on similarity between potential drivers of mammal abundance, including distance to settlements, roads, rivers, protected areas, elevation and time since logging (2-10 years before our study), although some camera grids overlapped older logging blocks. |
|---|---|
| Location | We set up arrays of camera traps from 2018 to 2021 in 14 logging concessions owned by 11 different companies (5 FSC and 6 non-FSC) in Gabon and the Republic of Congo. |
| Access & import/export | We gained permission from all participating logging companies to access their concessions. |
| Disturbance | Paths were made in the forest to be able to install the camera traps. Tropical forests are highly resilient such small disturbances as plants regrow quickly. |

# Reporting for specific materials, systems and methods

We require information from authors about some types of materials, experimental systems and methods used in many studies. Here, indicate whether each material, system or method listed is relevant to your study. If you are not sure if a list item applies to your research, read the appropriate section before selecting a response.

## Materials & experimental systems

| n/a | Involved in the study |
|---|---|
| ☒ | Antibodies |
| ☒ | Eukaryotic cell lines |
| ☒ | Palaeontology and archaeology |
| ☐ | ☒ Animals and other organisms |
| ☒ | Clinical data |
| ☒ | Dual use research of concern |
| ☒ | Plants |

## Methods

| n/a | Involved in the study |
|---|---|
| ☒ | ChIP-seq |
| ☒ | Flow cytometry |
| ☒ | MRI-based neuroimaging |

## Animals and other research organisms

Policy information about studies involving animals; ARRIVE guidelines recommended for reporting animal research, and Sex and Gender in Research

| | |
|---|---|
| Laboratory animals | The study did not involve laboratory animals. |
| Wild animals | We confirm that no wild animals were harmed in any way as animals were only observed using camera traps. |
| Reporting on sex | The study did not involve sex-based data or analysis. |
| Field-collected samples | The study did not involve samples collected from the field. |
| Ethics oversight | No ethical approval or guidance was required because animals were only remotely observed using camera traps. |

Note that full information on the approval of the study protocol must also be provided in the manuscript.

