## [Peer Review File · Nature]

Manuscript Title: FSC-certified forest management benefits large mammals compared to non-FSC

Reviewer Comments & Author Rebuttals

Reviewer Reports on the Initial Version:

Referees' comments:

Referee #1 (Remarks to the Author):

This manuscript shows that photographic encounter rates of mammals are lower (with particularly strong differences for large mammals and mammals of high conservation concern) in paired non-FSC certified forestry concessions compared to FSC certified concessions in Gabon and the Republic of Congo.

My main conceptual concern is that this study reduce the difference between FSC and non-FSC sites down to the prevalence of hunting, when in fact, FSC certification requires much more than implementing management to reduce illegal hunting. The list of 10 key FSC criteria provided by the authors shows that FSC certification requires a holistic approach to forest management, of which controlling illegal hunting is but one minor component. Many other criteria, such as maintaining high-value forest or including local communities into management (the authors mention regulated legal subsistence hunting in their FSC concessions) likely have positive effects on wildlife. Especially the use of reduced impact logging practices (over conventional selective logging) is likely to have major effects on levels of forest disturbance, which affects many tropical mammals (e.g., Tilker et al. 2019 www.nature.com/articles/s42003-019-0640-y) – the benefits of reduced impact logging (over conventional logging) for biodiversity including mammals have been reviewed repeatedly (e.g., Bicknell et al. 2015 <https://doi.org/10.1111/1365-2664.12391>). Yet, there is no consideration in the entire paper of potential differences in habitat disturbance and disturbance histories. Even if logging histories cannot be obtained, habitat disturbance could have been assessed in situ, and could potentially still be assessed using remote sensing data (eg NDMI, though that may not work well as an indicator of forest disturbance in humid tropical forests; or aboveground carbon). In short, it seems unlikely to me that the differences between FSC and non-FSC concessions can be reduced down to solely the level of hunting pressure, as this study implies; and this is thus primarily a study of the effect of hunting – not FSC certification – on mammals.

My second conceptual concern relates to the fact that while the 10 guiding principles of FSC certification are global, their specific implementation is decided on the national level. I appreciate that the present study is a large scale study, particularly for tropical systems, but I don't know how globally applicable the results are and that limitation is not discussed.

Another major concern relates to the use of encounter rates as an abundance measure. Encounter rates should not be interpreted as a measure of abundance. It is a mixed measure of abundance and activity, and without additional information, we cannot disentangle if increases in encounter rate are due to increases in abundance or activity (e.g., a camera trap that happens to be near a den will have much higher encounter rates than a camera not near a den, even if the same number of individuals occur in the general vicinity of the camera trap). The paper generally refer to the response variable as 'encounter rate', which is appropriate. But in several instances, it is characterized as a proxy for abundance and therefore superior to, for example, similar studies that evaluated diversity. This interpretation of encounter rate as abundance is especially evident in the calculation of a biomass value. Interpretation of the analysis of encounter rates should always treat them as the mix they are and always acknowledge that we simply don't know if differences in encounter rates are due to differences in abundance or behavior (or most likely both). While the authors are correct that it is less problematic to compare encounter rates of the same species among sites, it is nonetheless potentially problematic. Species can change their home ranges and movement patterns in response to disturbance, which can affect encounter rates. I do not object

to looking at encounter rates to study differences in wildlife among sites; but I do object to interpreting results as differences in abundance.

I also have some reservations about the analytical choice, though I acknowledge this may be a question of style. I think a parametric hierarchical model (to account for the nestedness of camera traps within sites) that can take into account potential confounding variables such as elevation, distance to towns, location on a trail etc, would be a more powerful tool for the purpose of this study. There were no significant differences in medians between FSC and non-FSC sites with respect to several covariates that may affect either abundance or activity (ie, encounter rates), but for some, the ranges are clearly very different. Responses to covariates are not always linear along the entire covariate gradient (e.g., many species show a quadratic response to elevation), so the fact that the medians are similar does not necessarily mean that there is no confounding influence of some of these covariates on median encounter rates. Such an approach would also have allowed you to group species into weight, taxonomic etc classes (which, technically, you could also do in an occupancy framework).

Finally, I am not sure how accurate the method for determining hunting pressure is – looking for signs on the last 500m on the way to a camera location, rather than doing systematic sign searches around each camera location. Several studies have shown that signs of hunting may not be reliably detected (eg., Ibbett et al. 2020 <https://doi.org/10.1016/j.biocon.2020.108581>), this is likely a more severe problem when no dedicated search effort for hunting signs is implemented. What that means for your study should be discussed.

Additional in-line comments

L60: Mammal encounter rate is no more an abundance measure than occupancy or diversity

Ex.D. Fig 5: While the medians aren't significantly different, the spread for some of these variables seems to be considerably different between FSC and non-FSC concessions.

L255: That means a maximum of 5x5 km, or 25km² were sampled. How much of a given concession's area is that? Is one small(ish) grid representative of the larger concession?

L262: Forest visibility measured how?

L265: Hunting signs – presence or some form of count?

L276f: You cannot ascertain that these were indeed different individuals. The most you can do is treat them as independent detections, with no information of that is of the same or of multiple individuals. Even if you rarely obtain multiple photos of the same species in a short period of time (as your sensitivity analysis suggests), that does not mean that the photos at a given site are not potentially of the same individual.

L283ff: This is not a very convincing test of equality in detection probability. Detection probability/rates are affected by many aspects of behavior, not just curiosity or shyness toward a new object. These quantities are affected by movement rates and overall diurnal activity rates, which may be affected by hunting. The fact that there was no change in detection rate over time does not indicate that detection probability is the same for the two types of concession.

L304: Why would there be country specific effects on biomass but nothing else?

L306ff: Not quite clear what you analyze here – do you compare encounter rates (or rather, relative differences in rates) between weight classes? Why, what is your hypothesis? So, the main effect relates to weight class, taxonomic group etc?

L314ff: Not clear what these add and what you compare

L86: The figures (or the results text) don't give the Friedman test results you mention in the methods.

Referee #2 (Remarks to the Author):

Very good study indeed. Needs to be generalized beyond west Africa - should be doable.

Referee #3 (Remarks to the Author):

Please see attached document

Referee #5 (Remarks to the Author):

The manuscript by Zwerts et al. assesses the effect of FSC certification on mammal abundance, diversity and biomass. The authors do this by collecting camera trap data in FSC certified and non-certified concessions in the Congo Basin. The results show that mammal encounter rate varies with certification status, with large mammals, critically endangered species and certain taxonomic groups being more often encountered in certified concessions than in non-certified ones. These results have large implications as they convincingly showed that timber extraction does not lead to overhunting when the right management practices (in this case following the FSC regulations) are applied, supporting the idea that forest concessions can play a pivotal role for mammal conservation.

The strong points of the manuscript are that 1) the questions being addressed are extremely relevant as the role of (certified) forest concessions/ forest management for biodiversity conservation is a contested issue in the conservation literature, 2) it provides empirical data that allows assessing if FSC certification reduces hunting pressure or not, 3) the study has a robust experimental design that includes multiple sites (14 concessions) and allows controlling for differences among sites (by using a paired site approach), 4) it is large-scale study, focussing a broad range of mammals, and using a large number of observations.

The points that can be improved on this manuscript are the following: 1) conservation implications of the results should be expanded (given the abovementioned strong points), 2) more-in-depth discussion on the practices needed to move towards a well-managed company with low negative impact on wildlife, 3) consideration of using a different statistical analysis, 4) the effect of time since logging is not explored in the analysis.

MAJOR COMMENTS

1. Conservation implications should be expanded given the importance of your findings for biodiversity conservation in Central Africa or elsewhere in the tropics. Several authors have claimed (without much evidence) that logging cannot be seen as a conservation tool because it opens the area to hunters, resulting in an increase in hunting pressure and further defaunation of the area. Your work shows very clearly (and with evidence!) that certified concessions do better than non-certified concessions in terms of mammal abundance, probably due to all the regulations and requirements that need to be followed by certified companies (e.g., closing road, patrols). Therefore, I think that you should expand the section on conservation implications of your discussion with the following aspects:

- Provide a larger view on the implications of your results than you do at this moment (lines 198-210). How do your results compare with other studies working on protected areas? And how do your results compare with studies done in areas that are not being managed for protection or for timber production? In other words, what is the contribution for biodiversity conservation for (non) certified companies when you compared them to protected areas and to forests without a clear manager? I can imagine that this is difficult to do as you do not have data on this, but these are issues and questions that have been in the literature without much empirical support, and I think that you have at least part of the data to explore these issues (with caution, obviously).
- The potential role of well-managed concessions for landscape connectivity among protected areas. You mention this very briefly (line 207) but given the large distribution ranges of some of the species, I think that it is worthwhile to expand this part, and to discuss the role of managed forests in that regard.
- Put the size of the concessions into perspective as the concessions in your region are large and many readers will not be aware of their size. By doing this it becomes clear that concessions occupy large areas, and that together with protected areas can have a large contribution on

wildlife conservation.

2. More-in-depth discussion on the practices needed to move a non-certified company towards a well-managed company. You certainly discussed some of these aspects in the section "The necessity to upscale certification", but I would like to see more specific recommendations in that regard. What are the basic practices listed among the Principal, Criteria and Indicators would you advice to start with? I miss that kind of recommendations, which are important as such recommendation can be included in national codes of practices and forest laws.

3. Analysis was done using Wilcoxon signed-rank tests which is a very conservative test. I agree with the analysis done by the authors, but it is also a rather conservative analysis. I think that the authors could have done a mixed generalized linear model, using the data per camera trap. The model would be the following: camera trap nested within concession, paired concession as random factor, the fixed factor certification status (yes/no), and the different variables they measured. I think that in this way the authors could explore more the effect of certification status because their sample size would be much larger (28-36 camera x 14 concession) than the 14 data points they have at this moment. I think that the results would not change in direction but it is possible that stronger effects of certification would be found (as the test currently used is very conservative). Then the authors could run a similar analysis, with all other potential drivers of animal abundance (distance to settlements, roads, rivers, protected areas, elevation, time since logging, etc.) as fixed factors. In that way, they could assess how the different factors explain the response variables assessed.

4. Effect of time since logging is not tested even though we now that forest structure changes over time, and that these changes may vary between certified and non-certified companies (as damage is larger in the later). The results of such test should be included in Extended Data Table 6.

MINOR COMMENTS

- Line 37-38: is one of the measures taken by certified companies the provision of alternative sources of meat? In the certified companies I have worked, the company would offer cow meat to workers (instead of a gun and bullets) to reduce the need to hunt. Is that the case also in your area? If so, please mention it.

-Line 49-51: please revise the sentence. Negative effects of timber extraction ON WHAT?

-Line 49-57: please revise the text to clarify better the situation of the concession in this region. The text now gives the idea that forest are intact and that the only disturbance is logging but then you say that hunting is pervasive. So logging is not the only disturbance, is it?

-Line 71-72: provide more information about the study design in the legend of the figure, such as number camera per site, distance between camera. I think that this is important because the methods is provided much later.

-Line: 198: strictly speaking the term "conservation" refers to the protection and sustainable use of a given area. So does the 21.1% you refer to include also concessions or is the value restricted to protected areas? If the latter is the case, then I would be explicit about it and talk about protection rather than conservation.

-Line 198-200: I struggle with this sentence because it goes from % of conservation area to % of species distribution area found in concessions.

-Line 200-201: this sentence is kind of lost in the arguments being presented in this paragraph. I think that you can elaborate much more than what you have done now. Please my first major comment above.

-Line 211: please rephrase as "FSC effectively controls" is not correct. FSC does not control itself, the rules/requirements/regulations of "FSC certification lead to a more effective control".

-Line 255: Indicate how many cameras were then put over time in the 14 concessions, and which area was covered by the cameras per concession.

-Line 271-275: what proportion of pictures were then not used because the species could not be identified or the rate observations?

Referee 3 comments:

FSC-certified forestry benefits large mammals compared to non-FSC

General Comments:

This paper contains a unique set of data that in general does show that there are significant differences in the abundance of mammal species in FSC-certified versus non-FSC certified concessions. I do think the paper merits publication but, in my opinion, it needs tightening up much more – I have tried to help with my comments below.

I believe there could be more detailed analyses of the possible effects of location, surrounding population densities and surrogates of anthropogenic factors (roads etc.) on the abundance measures calculated. The effects of certification on mammal abundance are obvious but we would learn a lot more if we knew whether certification is the main reason for the differences and not other factors.

Specific Comments:

Lines 31-32: Perhaps be more explicit about the fact that the studies cited here are only for Neotropical forests. So, no studies in Africa available.

Line 33: The statement “ Yet, little is known...” could be better qualified. For example, what do you mean by the “impact” of FSC certification on wildlife. From the previous sentence we know that FSC certified forests at least in Latin America have positive effects on large mammals such as jaguars and tapirs, but are you suggesting that there is little data on the status of faunal communities in FSC versus non-FSC certified forests? If so, make this clear.

Line 37: The tendency now is to refer to wild meat rather than bushmeat. See Coad et al. 2019 for the arguments for this.

Line 45: The term “eco-guards” may not be well known to many readers. Change to wardens or the like.

Lines 34-55: I think this paragraph and Lines 31-33 can be better integrated and shortened. I would describe succinctly what is known about the effects of FSC, highlight the shortcomings of previous studies (such as the lack of multiple sites etc.).

Line 53-55: This sentence is a bit confusing. This, I take it, says that abundance (population numbers rather than occupancy) is important rather than just diversity measures. This point needs to be much more clearly emphasised here.

Line 58: Explain which mammals. Large or medium-sized and large. What body mass range? I see from the lines further down that you classified mammal species by the weight ranges. In this line, indicate that all mammals (less than ??kg) were used in the study (certainly because of the limitations of the camera traps).

Line 64-65: Not quite sure what you mean by the sentence “We included all but one...” Do you mean that you covered all FSC-certified companies except one in the region? Make this clearer.

Line 67: No need to repeat “to assess the impact of FSC-certified logging”.

Line 71: Rather the references included here i.e., Potapov et al (2017) and Wilkie et al. (2011) which discuss the empty forest syndrome, I think defaunation papers are better suited as references. I would cite the relatively new paper Bugir et al. (2021) in *Food Webs* 26 (2021) e00183 where they clearly show that “Hunter-gatherers preferentially hunted 11 large-bodied, riskier species, and were capable of capturing species ranging from 0.6 to 535.3 kg but avoided those smaller than 2.5 kg”.

Line 86: Where is the species diversity data shown? I think there should be a third graph in this figure showing species diversity differences between FSC- certified concessions and non-FSC certified concessions.

Line 97: What is the accuracy of the camera traps in picking up smaller rodents, e.g., mice. Larger pouched rats and squirrels are going to be the main animals caught by the camera traps. What are the smallest animals (in g?) caught on cameras? It is unlikely that these are going to be adequately represented in your samples.

Lines 138-152: The authors should be careful in not repeating the results of the study in the discussion. I think this first paragraph should be much more general. Also, Lines 151-152 opens a very interesting line to explain the differences between the two types of concessions. I think the references included here are too general. There is a lot more on the issue of density compensation in hunted versus non-hunted forests that could be discussed and perhaps cite studies like Carlos Peres and Paul Dolman’s in *Oecologia* (2000) 122:175–189. Although there have been some attempts to measure density compensation in African forest faunas, I think somewhat unsuccessfully e.g., Josh Linder & John Oates in *Biological Conservation* (2011) 44, 738-745, the authors can do a lot more to advance this effect in their study.

Lines 153-163: The refuge effect of FSC-certified concessions is likely to be true, but this is likely to vary depending on the characteristics of FSC-certified concessions (and the non-FSC concessions also). For example, does distance to continuous forest or the presence of more human settlements or infrastructure affect the abundance of the different species. Significant differences can be observed between paired concessions. How can you explain these contrasts? I think it would add to the paper if the explanation of the differences between concession types can be more nuanced.

Lines 164-166: Your explanation of the lack of differences in species diversity is suggesting that there has not been any loss of species in any concession. Is this true for all sites? Also, the assertion that human population density is low around all sites, but it is not the same in all areas. I think some understanding of how human numbers (actually, a good surrogate of hunter density) and anthropogenic factors correlate with the results of the study is important.

Line 167: Better if the sentence states the importance of large mammals in the functioning of tropical forests because they are seed dispersers etc.

Line 171: This sentence could be better described – Forest carbon storage is greater where there are more intact large mammal assemblages, because they allow large trees to flourish etc. as shown in Peres et al. (2016) *Proc. Natl. Acad. Sci.* 113, 892–897.

Lines 171-175: The assertion in the sentence “In addition, by being more biodiverse....” is unknown yet. There are many conditions and variables that influence zoonotic spread. It would be interesting to find out whether disease is less rife around better protected areas.

Line 185-186: The sentence “the differences between FSC-certified and non-FSC concessions that may drive wildlife abundance, were non-significant” is not substantiated in the paper since there are no data on the characteristics of each concession and the differences between them. This should be included as indicated above.

Author Rebuttals to Initial Comments:

Response to reviewers

Line numbers refer to the manuscript with track changes.

Referees' comments:

Referee #1 (Remarks to the Author):

This manuscript shows that photographic encounter rates of mammals are lower (with particularly strong differences for large mammals and mammals of high conservation concern) in paired non-FSC certified forestry concessions compared to FSC certified concessions in Gabon and the Republic of Congo.

My main conceptual concern is that this study reduce the difference between FSC and non-FSC sites down to the prevalence of hunting, when in fact, FSC certification requires much more than implementing management to reduce illegal hunting. The list of 10 key FSC criteria provided by the authors shows that FSC certification requires a holistic approach to forest management, of which controlling illegal hunting is but one minor component. Many other criteria, such as maintaining high-value forest or including local communities into management (the authors mention regulated legal subsistence hunting in their FSC concessions) likely have positive effects on wildlife. Especially the use of reduced impact logging practices (over conventional selective logging) is likely to have major effects on levels of forest disturbance, which affects many tropical mammals (e.g., Tilker et al. 2019 www.nature.com/articles/s42003-019-0640-y) – the benefits of reduced impact logging (over conventional logging) for biodiversity including mammals have been reviewed repeatedly (e.g., Bicknell et al. 2015 <https://doi.org/10.1111/1365-2664.12391>). Yet, there is no consideration in the entire paper of potential differences in habitat disturbance and disturbance histories. Even if logging histories cannot be obtained, habitat disturbance could have been assessed in situ, and could potentially still be assessed using remote sensing data (eg NDMI, though that may not work well as an indicator of forest disturbance in humid tropical forests; or aboveground carbon). In short, it seems unlikely to me that the differences between FSC and non-FSC concessions can be reduced down to solely the level of hunting pressure, as this study implies; and this is thus primarily a study of the effect of hunting – not FSC certification – on mammals.

We thank the reviewer for the detailed comment and agree that FSC certification does not only exert its influence through the reduction of hunting. We included a section in the discussion explaining that the FSC approach is holistic, that hunting controls are vital - in line with Tilker et al. (2019) www.nature.com/articles/s42003-019-0640-y - but that other factors such as retention of high conservation value areas and reduced impact logging practices are likely to contribute to the observed effects as well.

Lines 231 - 241:

“The FSC approach is holistic and controlling hunting is likely the most important factor for the reduction of environmental impacts in logged tropical forests⁷. We found more hunting signs in non-FSC concessions, which supports the interpretation that FSC effectively reduces hunting pressure, although we recognize that counting hunting signs are a relatively weak measure for the quantification of hunting pressure⁴⁹. Hunting has long been known to be the most important driver of forest fauna decline in Central African logged forests^{8,50} and the same phenomenon has recently been shown in Asia⁷. Nonetheless, other factors such as retaining high conservation value areas and reduced impact logging practices are likely to contribute to the observed effects as well⁵¹. Yet, our data do not allow for causal inference of the association of any of the specific measures

implemented by FSC companies with the observed effects as that would require setting up more detailed measure-based experiments.”

We do however think that hunting is a major indirect impact of logging (forest mammals), as hunting has a much more lasting impact on mammal populations than the selective removal of trees and the creation of forest roads. This is also confirmed by a recent paper which showed that of the four main threats to biodiversity in sub-Saharan Africa, hunting is the most important one in the Central African region (Leisher et al., 2022 ‘Ranking the direct threats to biodiversity in sub-Saharan Africa’).

Moreover, we also read the article referred to by the reviewer (Tilker et al. 2019 www.nature.com/articles/s42003-019-0640-y), and in fact wish to use it to reinforce our argument about the importance of managing hunting in logging concessions given the following statement in Tilker et al. (2019):

“Our findings suggest that intensive, indiscriminate hunting may be a more immediate threat than moderate habitat degradation for tropical faunal communities, and that conservation stakeholders should focus as much on overhunting as on habitat conservation to address the defaunation crisis”

We indeed do not include a remotely sensed measure of habitat disturbance as we do not believe that such measures will effectively quantify the level of below canopy disturbance of lightly logged forests. We do however include visual estimations of visibility and other site covariates in Figures S2, S3 and S4 which were similar between FSC and non-FSC concessions and these covariates did not improve linear mixed models. Given the literature about the relative importance of habitat disturbance and our consideration of site covariates in our research design and analyses, we hope the reviewer agrees that our focus on hunting as a proxy for environmental effectiveness of FSC is indeed justified.

To better clarify the relative importance of direct and indirect impacts on forest biodiversity we have rephrased the section as follows:

Lines 50 - 61:

“The FSC aims to reduce direct environmental impacts through, among others, maintaining high conservation values forests and applying reduced impact logging practices (Tables S1 & S2). A major concern for biodiversity is that timber extraction – by the creation of roads – creates access to previously remote forests, which facilitates illegal and unsustainable hunting⁴⁻⁸. This indirect effect of logging mainly influences medium to large-sized forest mammals, which are particularly vulnerable to human encroachment^{9,10}. FSC certification may diminish these negative impacts because, among other measures, companies have to reduce accessibility to concessions by closing off old logging roads, prohibit wild meat transport or hunting materials, provide alternative sources of proteins and carry out surveillance by rangers. An FSC certificate is valid for five years and logging companies are third party audited for compliance on a yearly basis using annual surveillance assessments.”

My second conceptual concern relates to the fact that while the 10 guiding principles of FSC certification are global, their specific implementation is decided on the national level. I appreciate that the present study is a large scale study, particularly for tropical systems, but I don’t know how globally applicable the results are and that limitation is not discussed.

We thank the reviewer for the comment and have now included a more detailed discussion about extrapolation of our results to other areas.

Lines 303 - 309:

“The results of this study are likely to be applicable to other logged tropical forests where hunting through increased accessibility poses a risk to forest wildlife, as the measures to reduce impact on wildlife are also applied in other FSC-certified tropical forestry systems. We infer this with caution as timber extraction volumes, concession size and shape, presence of public roads, population density and other characteristics may differ between concessions and thereby affect the impacts of FSC-certified forest management⁶⁴”

Another major concern relates to the use of encounter rates as an abundance measure. Encounter rates should not be interpreted as a measure of abundance. It is a mixed measure of abundance and activity, and without additional information, we cannot disentangle if increases in encounter rate are due to increases in abundance or activity (e.g., a camera trap that happens to be near a den will have much higher encounter rates than a camera not near a den, even if the same number of individuals occur in the general vicinity of the camera trap). The paper generally refer to the response variable as ‘encounter rate’, which is appropriate. But in several instances, it is characterized as a proxy for abundance and therefore superior to, for example, similar studies that evaluated diversity. This interpretation of encounter rate as abundance is especially evident in the calculation of a biomass value. Interpretation of the analysis of encounter rates should always treat them as the mix they are and always acknowledge that we simply don’t know if differences in encounter rates are due to differences in abundance or behavior (or most likely both). While the authors are correct that it is less problematic to compare encounter rates of the same species among sites, it is nonetheless potentially problematic. Species can change their home ranges and movement patterns in response to disturbance, which can affect encounter rates. I do not object to looking at encounter rates to study differences in wildlife among sites; but I do object to interpreting results as differences in abundance.

We thank the reviewer for the comment and understand the concern. We removed the sentence where encounter rates are characterized as a proxy for abundance.

Lines 258 - 261:

“We used encounter rate, defined as the number of observations divided by the number of camera trap days. Encounter rates may be affected by unaccounted influences on detection probabilities⁵⁴, which may complicate comparisons between species, or between sites.”

We have also included a more detailed discussion about the use of encounter rates in relation to our results:

Lines 276 - 281:

“It is however important to note that encounter rates are a mixed measure of abundance and activity and we cannot disentangle whether changes in encounter rate are due to changes in abundance, activity, or both. Species’ home ranges and movement patterns can change in response to disturbance, which can affect encounter rates. We also estimate biomass using encounter rates, which is a useful proxy to assess differences between forest management types, but cannot be interpreted as true biomass.”

I also have some reservations about the analytical choice, though I acknowledge this may be a question of style. I think a parametric hierarchical model (to account for the nestedness of camera traps within sites) that can take into account potential confounding variables such as elevation, distance to towns, location on a trail etc, would be a more powerful tool for the purpose of this study. There were no significant differences in medians between FSC and non-FSC sites with respect to several covariates that may affect either abundance or activity (ie, encounter rates), but for some, the ranges are clearly very different. Responses to covariates are not always linear along the entire

covariate gradient (e.g., many species show a quadratic response to elevation), so the fact that the medians are similar does not necessarily mean that there is no confounding influence of some of these covariates on median encounter rates.

We thank the reviewer for the suggestion to use more sophisticated models despite the lack of significant differences of the covariates. We have now used linear mixed models, which yielded higher statistical power and allowed for modelling the influence of the covariates. To account for potential non-linear effects along the covariate gradients we tested quadratic terms for all covariates which did not result in improved models.

The analytical approach is described in:

Lines 433 - 446:

“To assess whether encounter rates varied between FSC and non-FSC concessions, we quantified the means of the paired concessions using linear mixed-effects models with concessions, concession pairs and cameras as random effects. We allowed the means of concession pairs to vary between weight class, taxonomic group and IUCN Red List category if supported by model selection. We tested whether potential drivers of wildlife abundance (Figs. S1, S2, S3 & S4) variables were important using a model-selection approach based on minimization of Bayesian Information Criterion (BIC) values. We found that the inclusion of covariates did not substantially improve the model for weight classes and IUCN categories. For all mammals pooled together and for taxonomic groups however, the inclusion of elevation, which was correlated to distance to roads and rivers, slightly improved the model. There was however very little variation between encounter rates and elevation and all camera trap locations were non-mountainous lowland rainforests, so whatever variation is linked to elevation is therefore assumed to be of very small importance (Fig. S6). Quadratic covariate terms did not result in better models. Post hoc comparisons were multivariate t adjusted.”

And discussed in:

Lines 246 - 249:

“While we believe that these covariates are important drivers of wildlife abundance⁵², including these covariates did not greatly improve the model, which underscores that camera grid locations were sufficiently similar in terms of these confounding influences.”

As the new modelling approach resulted in higher statistical power, Extended data figure 2 became redundant and is removed as the encounter rates of all subgroups included in this figure now differ significantly between the FSC and non-FSC groups.

Such an approach would also have allowed you to group species into weight, taxonomic etc classes (which, technically, you could also do in an occupancy framework).

We have removed the text about grouping into weight classes in relation to occupancy modelling:

Lines 258 – 262:

“We used encounter rate, defined as the number of observations divided by the number of camera trap days. Encounter rates may be affected by unaccounted influences on detection probabilities⁵⁴, which may complicate comparisons between species, or between sites.”

Finally, I am not sure how accurate the method for determining hunting pressure is – looking for signs on the last 500m on the way to a camera location, rather than doing systematic sign searches

around each camera location. Several studies have shown that signs of hunting may not be reliably detected (eg., Ibbett et al. 2020 <https://doi.org/10.1016/j.biocon.2020.108581>), this is likely a more severe problem when no dedicated search effort for hunting signs is implemented. What that means for your study should be discussed.

We agree with the comments and have addressed this result in the discussion by presenting it with caution. We chose to retain the data as it does support the interpretation that curbing hunting pressure is the major driver for effective wildlife conservation in tropical forest management.

Lines 231 - 235:

“We found more hunting signs in non-FSC concessions, which supports the interpretation that FSC effectively reduces hunting pressure, although we recognize that counting hunting signs are a relatively weak measure for the quantification of hunting pressure⁴⁹.”

Additional in-line comments

L60: Mammal encounter rate is no more an abundance measure than occupancy or diversity

We agree that encounter rates do not equal abundance and have rephrased the sentence accordingly to highlight the shortcoming of the discussed study:

Line 72 - 78:

“However, that study investigated species richness of birds, which have a relatively large dispersal ability and may not represent a strong indicator of localized management impacts. Moreover, in addition to comparing only the number of species, it is important to compare population sizes. Hunting not necessarily completely extirpates wildlife species, especially when forests are connected, but rather results in population declines⁴.”

Ex.D. Fig 5: While the medians aren’t significantly different, the spread for some of these variables seems to be considerably different between FSC and non-FSC concessions.

We have now adapted our analysis to include the potentially confounding effect of these variables, and BIC model selection showed that these covariates did not influence encounter rates substantially. We keep the figure to show that sites were carefully selected to avoid the influence of other drivers of abundance, which is explained in the methods:

Lines 357 – 360:

“Camera trap grid locations within each pair of concessions were chosen based on similarity between potential drivers of animal abundance, including distance to settlements, roads, rivers, protected areas, elevation (Figs. S1; Table S5)”

L255: That means a maximum of 5x5 km, or 25km² were sampled. How much of a given concession’s area is that? Is one small(ish) grid representative of the larger concession?

We agree that the sampling grids were small relative to the size of some of the concessions, but given the equality of other drivers of animal abundances between FSC and non-FSC sites and the replication of samples, we think we have sufficiently ensured comparability.

This is detailed in lines 242 - 246:

“For the sections of the concessions that we sampled, we ensured comparability between concessions by maximizing similarity in geographic covariates that may drive variation in wildlife

abundance, i.e. elevation and distances to roads, rivers, human settlements and protected areas, between each pair of FSC-certified and non-FSC concessions (Fig. S1; Table S5).”

L262: Forest visibility measured how?

It was a visual estimation. We have adapted the text to clarify this:

Lines 376 - 377:

“We also visually estimated forest visibility (0-10m / 11-20m / >20m), slope (0-5° / 5-20° / >20°),”

L265: Hunting signs – presence or some form of count?

We counted the hunting signs and have adapted the text accordingly:

Lines 378 – 380:

“When approaching each predefined camera point, we counted cartridges, snares and hunting camps from 500 m before the camera up to its location.”

In figure 2B, we display the proportion of camera locations with hunting signs of all camera locations and not the count data as hunting signs tend to be clustered based on recent hunting activity, resulting in a very high number of detections when hunting signs were indeed present.

L276f: You cannot ascertain that these were indeed different individuals. The most you can do is treat them as independent detections, with no information of that is of the same or of multiple individuals. Even if you rarely obtain multiple photos of the same species in a short period of time (as your sensitivity analysis suggests), that does not mean that the photos at a given site are not potentially of the same individual.

We agree with the reviewer that we cannot ascertain that detections separated 10 minutes in time were indeed different individuals and have rephrased the text accordingly:

Lines 390 - 397:

“Observations of the same species that were at least 10 minutes apart were considered as separate detections. We assessed the influence of this threshold with a sensitivity analysis by calculating the number of detections for intervals of 10, 30, 60 and 1440 minutes, which all yielded similar results (Table S6).”

L283ff: This is not a very convincing test of equality in detection probability. Detection probability/rates are affected by many aspects of behavior, not just curiosity or shyness toward a new object. These quantities are affected by movement rates and overall diurnal activity rates, which may be affected by hunting. The fact that there was no change in detection rate over time does not indicate that detection probability is the same for the two types of concession.

We agree with the reviewer that there may indeed be other factors that affect detection probability and have added this to the text:

Lines 406 - 408:

“We recognize however that other factors may exist that may have influenced detection probability, such as movement rates and diurnal activity rates, both of which may be affected by hunting.”

In line with this, we also recognize the reviewers earlier comment that “encounter rates are a mixed measure of abundance and activity, and without additional information, we cannot disentangle if increases in encounter rate are due to increases in abundance or activity”.

We have included text related to this concern in lines 276 - 281:

“It is however important to note that encounter rates are a mixed measure of abundance and activity and we cannot disentangle whether changes in encounter rate are due to changes in abundance, activity, or both. Species’ home ranges and movement patterns can change in response to disturbance, which can affect encounter rates. We also estimate biomass using encounter rates, which is a useful proxy to assess differences between forest management types, but cannot be interpreted as true biomass.”

L304: Why would there be country specific effects on biomass but nothing else?

It is indeed correct that if country specific effects would occur, they would not be limited to biomass. We chose one comprehensive measure as an example of country-specific effects as our proxy for biomass is a function of our encounter rates. We agree that this analysis is unnecessarily confusing and have removed the country-specific analysis and graphs as these effects can also be deduced from figure 2A together with Table S4, which displays the concession numbers and countries where they are located.

L306ff: Not quite clear what you analyze here – do you compare encounter rates (or rather, relative differences in rates) between weight classes? Why, what is your hypothesis? So, the main effect relates to weight class, taxonomic group etc?

We have replaced this entire section with the following text:

Lines 433 - 446:

“To assess whether encounter rates varied between FSC and non-FSC concessions, we quantified the means of the paired concessions using linear mixed-effects models with concessions, concession pairs and cameras as random effects. We allowed the means of concession pairs to vary between weight class, taxonomic group and IUCN Red List category if supported by model selection. We tested whether potential drivers of wildlife abundance (Figs. S1, S2, S3 & S4) variables were important using a model-selection approach based on minimization of Bayesian Information Criterion (BIC) values. We found that the inclusion of covariates did not substantially improve the model for weight classes and IUCN categories. For all mammals pooled together and for taxonomic groups however, the inclusion of elevation, which was correlated to distance to roads and rivers, slightly improved the model. There was however very little variation between encounter rates and elevation and all camera trap locations were non-mountainous lowland rainforests, so whatever variation is linked to elevation is therefore assumed to be of very small importance (Fig. S6). Quadratic covariate terms did not result in better models. Post hoc comparisons were multivariate t adjusted.”

L314ff: Not clear what these add and what you compare

See previous answer.

L86: The figures (or the results text) don't give the Friedman test results you mention in the methods.

In our new analysis we do not use Friedman test results anymore, so we have removed the text about this test from the methods.

Referee #2 (Remarks to the Author):

Very good study indeed. Needs to be generalized beyond west Africa - should be doable.

We thank the reviewer for the positive comment. We agree that the results of this study are likely to be applicable to other logged tropical forests where hunting through increased accessibility poses a risk to forest wildlife. We have now mentioned this explicitly:

Lines 303 - 309:

“The results of this study are likely to be applicable to other logged tropical forests where hunting through increased accessibility poses a risk to forest wildlife, as the measures to reduce impact on wildlife are also applied in other FSC-certified tropical forestry systems. We infer this with caution as timber extraction volumes, concession size and shape, presence of public roads, population density and other characteristics may differ between concessions and thereby affect the impacts of FSC-certified forest management⁶⁴.”

Referee #3 (Remarks to the Author):

General Comments:

This paper contains a unique set of data that in general does show that there are significant differences in the abundance of mammal species in FSC-certified versus non FSC certified concessions. I do think the paper merits publication but, in my opinion, it needs tightening up much more – I have tried to help with my comments below.

I believe there could be more detailed analyses of the possible effects of location, surrounding population densities and surrogates of anthropogenic factors (roads etc.) on the abundance measures calculated. The effects of certification on mammal abundance are obvious but we would learn a lot more if we knew whether certification is the main reason for the differences and not other factors.

We thank the reviewer for the suggestion to quantify the impact of other factors than only certification. We have now used linear mixed models, which yielded higher statistical power and allowed for modelling the influence of the covariates.

The analytical approach is described in:

Lines 433 - 446:

“To assess whether encounter rates varied between FSC and non-FSC concessions, we quantified the means of the paired concessions using linear mixed-effects models with concessions, concession pairs and cameras as random effects. We allowed the means of concession pairs to vary between weight class, taxonomic group and IUCN Red List category if supported by model selection. We tested whether potential drivers of wildlife abundance (Figs. S1, S2, S3 & S4) variables were important using a model-selection approach based on minimization of Bayesian Information Criterion (BIC) values. We found that the inclusion of covariates did not substantially improve the model for weight classes and IUCN categories. For all mammals pooled together and for taxonomic groups however, the inclusion of elevation, which was correlated to distance to roads and rivers, slightly improved the model. There was however very little variation between encounter rates and elevation and all camera trap locations were non-mountainous lowland rainforests, so whatever variation is linked to elevation is therefore assumed to be of very small importance (Fig. S6). Quadratic covariate terms did not result in better models. Post hoc comparisons were multivariate t adjusted.”

And discussed in:

Lines 246 - 249:

“While we believe that these covariates are important drivers of wildlife abundance⁵², including these covariates did not greatly improve the model, which underscores that camera grid locations were sufficiently similar in terms of these confounding influences.”

Specific Comments:

Lines 31-32: Perhaps be more explicit about the fact that the studies cited here are only for Neotropical forests. So, no studies in Africa available.

We thank the reviewer for the comment and have included information about the location of the cited studies. We did this both for the social studies that were mentioned, which were both done in the Congo Basin, and for the studies that compared FSC certification to protected areas, which were both done in Latin America.

Lines 62 - 64:

“In African tropical forests, a positive influence of FSC certification has been demonstrated in relation to reduced deforestation¹¹, and for social aspects, such as working and living conditions of employees and benefit sharing with neighbouring institutions¹². Previous studies in Latin America suggest that mammal occupancy in FSC-certified sites is comparable to that of protected areas^{13,14}.”

Line 33: The statement “ Yet, little is known...” could be better qualified. For example, what do you mean by the “impact” of FSC certification on wildlife. From the previous sentence we know that FSC certified forests at least in Latin America have positive effects on large mammals such as jaguars and tapirs, but are you suggesting that there is little data on the status of faunal communities in FSC versus non-FSC certified forests? If so, make this clear.

We agree with the reviewer that the sentence can be improved and have done so accordingly.

Lines 66 - 67:

“Yet, there is very little data on the status of faunal communities in FSC-certified versus non-FSC forests^{2,3}.”

Line 37: The tendency now is to refer to wild meat rather than bushmeat. See Coad et al. 2019 for the arguments for this.

We thank the reviewer for the observation and have changed bushmeat to wild meat throughout the manuscript.

Line 45: The term “eco-guards” may not be well known to many readers. Change to wardens or the like.

We thank the reviewer for this observation and have changed the text accordingly.

Lines 55 - 58:

“FSC certification may diminish these negative impacts because, among other measures, companies have to reduce accessibility to concessions by closing off old logging roads, prohibit wild meat transport or hunting materials, provide alternative sources of proteins and carry out surveillance by rangers.”

Lines 34-55: I think this paragraph and Lines 31-33 can be better integrated and shortened. I would describe succinctly what is known about the effects of FSC, highlight the shortcomings of previous studies (such as the lack of multiple sites etc.).

We have integrated lines 31-33 better by moving them down in slightly adapted form, to serve as the introduction on the section about previous studies comparing FSC and non-FSC.

Lines 42 - 68:

“Commercial timber concessions cover over one quarter of the world’s remaining tropical forests¹. Forest certification systems like the Forest Stewardship Council (FSC) aim to have more positive socio-economic and environmental outcomes compared to conventional logging schemes. The FSC aims to reduce direct environmental impacts through, among others, maintaining high conservation values forests and applying reduced impact logging practices (Tables S1 & S2). A major concern for biodiversity is that timber extraction – by the creation of roads – creates access to previously remote forests, which facilitates illegal and unsustainable hunting⁴⁻⁸. This indirect effect of logging mainly influences medium to large-sized forest mammals, which are particularly vulnerable to human encroachment^{9,10}. FSC certification may diminish these negative impacts because, among other measures, companies have to reduce accessibility to concessions by closing off old logging roads, prohibit wild meat transport or hunting materials, provide alternative sources of proteins and carry out surveillance by rangers. An FSC certificate is valid for five years and logging companies are third party audited for compliance on a yearly basis using annual surveillance assessments.

In African tropical forests, a positive influence of FSC certification has been demonstrated in relation to reduced deforestation¹¹, and for social aspects, such as working and living conditions of employees and benefit sharing with neighbouring institutions¹². Previous studies in Latin America suggest that mammal occupancy in FSC-certified sites is comparable to that of protected areas^{13,14}. Yet, there is very little data on the status of faunal communities in FSC-certified versus non-FSC forests^{2,3}. Most studies on the effectiveness of FSC certification concerning wildlife conservation have focused on one or a few sites or species at a time¹⁵⁻¹⁸.”

Line 53-55: This sentence is a bit confusing. This, I take it, says that abundance (population numbers rather than occupancy) is important rather than just diversity measures. This point needs to be much more clearly emphasised here.

We have adapted the sentence and emphasized more clearly that comparing population sizes is more important than comparing only species numbers.

Lines 72 - 78:

“However, that study investigated species richness of birds, which have a relatively large dispersal ability and may not represent a strong indicator of localized management impacts. Moreover, in addition to comparing only the number of species, it is important to compare population sizes. Hunting not necessarily completely extirpates wildlife species, especially when forests are connected, but rather results in population declines⁴.”

Line 58: Explain which mammals. Large or medium-sized and large. What body mass range? I see from the lines further down that you classified mammal species by the weight ranges. In this line, indicate that all mammals (less than ??kg) were used in the study (certainly because of the limitations of the camera traps).

We added ‘small to large sized’ to the statement

Lines 81 - 84:

“We compared small to large-sized mammal observations across seven paired FSC-certified and non-FSC concessions in Gabon and the Republic of Congo, including all but one of the FSC-certified companies in Western Equatorial Africa (WEA) (Fig. 1).”

Line 64-65: Not quite sure what you mean by the sentence “We included all but one...” Do you mean that you covered all FSC-certified companies except one in the region? Make this clearer.

We have now made this clearer by adding the information to the previous statement about our sampling locations.

Lines 81 - 84:

“We compared small to large-sized mammal observations across seven paired FSC-certified and non-FSC concessions in Gabon and the Republic of Congo, including all but one of the FSC-certified companies in Western Equatorial Africa (WEA) (Fig. 1).”

Line 67: No need to repeat “to assess the impact of FSC-certified logging”.

We thank the reviewer for the observation and have adapted the sentence accordingly:

Lines 91 - 94:

“We calculated mammal encounter rate and grouped mammal species into five weight classes (Table S3), as the relative encounter rate of these classes can be used as a proxy for hunting pressure.”

Line 71: Rather the references included here i.e., Potapov et al (2017) and Wilkie et al. (2011) which discuss the empty forest syndrome, I think defaunation papers are better suited as references. I would cite the relatively new paper Bugir et al. (2021) in *Food Webs* 26 (2021) e00183 where they clearly show that “Hunter-gatherers preferentially hunted 11 large-bodied, riskier species, and were capable of capturing species ranging from 0.6 to 535.3 kg but avoided those smaller than 2.5 kg”.

We thank the reviewer for the suggestion and have replaced Wilkie et al. (2011) with Bugir et al. (2021) in line 95. We retained Potapov et al (2017) because it refers to a different statement.

Line 86: Where is the species diversity data shown? I think there should be a third graph in this figure showing species diversity differences between FSC- certified concessions and nonFSC certified concessions.

We did not find differences in species diversity between FSC and non-FSC concessions (Extended Data Table 1) and we therefore prefer to focus the results and this paper on encounter rates. This choice is also in line with our hypothesis that due to the relatively low timber extraction volumes in Western Equatorial Africa and the large extent of forests with potential buffer populations, we did not expect species to become completely extirpated.

This is also discussed in lines 109 - 112:

“We detected 55 mammal species and found a positive effect of FSC-certification on overall mammal encounter rate (Fig. 2A) and fewer signs of hunting (Fig. 2B) in FSC-certified than in non-FSC concessions (Extended Data Tables 1 & 2). We did not find marked differences in overall species diversity between the two concession types.”

Line 97: What is the accuracy of the camera traps in picking up smaller rodents, e.g., mice. Larger pouched rats and squirrels are going to be the main animals caught by the camera traps. What are the smallest animals (in g?) caught on cameras? It is unlikely that these are going to be adequately represented in your samples.

We thank the reviewer for the comment. It is indeed correct that small wildlife typically has a lower detection probability than for example elephants. Due to a difference in detection probability we do not report abundances, but merely encounter rates, which are among others indeed a function of detection probability. Nonetheless, the difference in detection probability between small and large wildlife is assumed to be similar between FSC and non-FSC concessions; therefore an interspecies representation difference should in our view not influence the weight class analysis.

Lines 138-152: The authors should be careful in not repeating the results of the study in the discussion. I think this first paragraph should be much more general.

Also, Lines 151-152 opens a very interesting line to explain the differences between the two types of concessions. I think the references included here are too general. There is a lot more on the issue of density compensation in hunted versus non-hunted forests that could be discussed and perhaps cite studies like Carlos Peres and Paul Dolman’s in *Oecologia* (2000) 122:175–189. Although there have been some attempts to measure density compensation in African forest faunas, I think somewhat unsuccessfully e.g., Josh Linder & John Oates in *Biological Conservation* (2011) 44, 738-745, the authors can do a lot more to advance this effect in their study.

We thank the reviewer for this comment and have deleted some repetition of results and added a section about density compensation with the suggested references. We deleted the Young et al., 2016 reference as it was indeed very general, but retained Yasuoka et al., 2015 as this study was done in the same region as our study.

Lines 193 - 194:

Deleted "Forest elephants, primates, large carnivores and medium to large forest antelopes were all encountered more frequently in FSC-certified concessions."

Lines 196 - 201:

Added "Non-FSC concessions contained proportionally more rodents and other small species than FSC-certified concessions (Extended Data Table 1). The lack of hunting impacts on small mammal populations suggests some form of density compensation is in place as the hunting pressure on small mammal populations might be compensated by higher reproductive rates, or a release from competition and predation in the non-FSC concessions³³⁻³⁵."

Lines 153-163: The refuge effect of FSC-certified concessions is likely to be true, but this is likely to vary depending on the characteristics of FSC-certified concessions (and the non-FSC concessions also). For example, does distance to continuous forest or the presence of more human settlements or infrastructure affect the abundance of the different species. Significant differences can be observed between paired concessions. How can you explain these contrasts? I think it would add to the paper if the explanation of the differences between concession types can be more nuanced.

We agree with the reviewer that more explanatory information would add to the paper and have now included the relation of covariates with encounter rates by running a new model. We have now used linear mixed models, which yielded higher statistical power and allowed for modelling the influence of the covariates. To account for potential non-linear effects along the covariate gradients we tested quadratic terms for all covariates which did not result in improved models.

The analytical approach is described in:

Lines 433 - 446:

"To assess whether encounter rates varied between FSC and non-FSC concessions, we quantified the means of the paired concessions using linear mixed-effects models with concessions, concession pairs and cameras as random effects. We allowed the means of concession pairs to vary between weight class, taxonomic group and IUCN Red List category if supported by model selection. We tested whether potential drivers of wildlife abundance (Figs. S1, S2, S3 & S4) variables were important using a model-selection approach based on minimization of Bayesian Information Criterion (BIC) values. We found that the inclusion of covariates did not substantially improve the model for weight classes and IUCN categories. For all mammals pooled together and for taxonomic groups however, the inclusion of elevation, which was correlated to distance to roads and rivers, slightly improved the model. There was however very little variation between encounter rates and elevation and all camera trap locations were non-mountainous lowland rainforests, so whatever variation is linked to elevation is therefore assumed to be of very small importance (Fig. S6). Quadratic covariate terms did not result in better models. Post hoc comparisons were multivariate t adjusted."

And discussed in:

Lines 246 - 249:

“While we believe that these covariates are important drivers of wildlife abundance⁵², including these covariates did not greatly improve the model, which underscores that camera grid locations were sufficiently similar in terms of these confounding influences.”

As the new modelling approach resulted in higher statistical power, Extended data figure 2 became redundant and is removed as all subgroups that were represented in this figure now have significant differences between the FSC and non-FSC groups.

Lines 164-166: Your explanation of the lack of differences in species diversity is suggesting that there has not been any loss of species in any concession. Is this true for all sites?

Some species were observed in FSC concessions but not in non-FSC concessions (see table below)

	Cluster 1	Cluster 2	Cluster 3	Cluster 4	Cluster 5	Cluster 6	Cluster 7
V1	Civettictis civetta	Euoticus elegantulus	Cephalophus ogilbyi	Hyemoschus aquaticus	Caracal aurata	Panthera pardus	Cephalophus ogilbyi
V2	Cercopithecus nictitans	Cephalophus nigrifrons	Euoticus elegantulus	Gorilla gorilla	Civettictis civetta	Caracal aurata	Loxodonta cyclotis
V3	Orycteropus afer	Dendrohyrax dorsalis	Cephalophus nigrifrons	Civettictis civetta	Cercopithecus nictitans	Smutsia gigantea	Cephalophus leucogaster
V4	Euoticus elegantulus	Poiana richardsonii		Cercopithecus nictitans		Orycteropus afer	Smutsia gigantea
V5	Mellivora capensis			Cephalophus nigrifrons		Syncerus caffer	Syncerus caffer
V6				Dendrohyrax dorsalis		Mellivora capensis	Mellivora capensis
V7				Poiana richardsonii		Cephalophus nigrifrons	Aonyx congicus
V8						Arctocebus aureus	
V9						Tragelaphus eurycerus	

But the other way around, some species were observed in non-FSC sites but not in FSC concessions (see table below)

	Cluster 1	Cluster 2	Cluster 3	Cluster 4	Cluster 5	Cluster 6	Cluster 7
V1	Phataginus tetradactyla	Syncerus caffer	Nandinia binotata	Cercopithecus cephus	Cercopithecus cephus	Cercopithecus indet	Cercopithecus cephus
V2		Phataginus tetradactyla	Cercopithecus cephus	Syncerus caffer	Syncerus caffer	Hylochoerus meinertzhageni	Mandrillus sphinx
V3		Galagoideus thomasi	Crossarchus platycephalus	Euoticus elegantulus	Phataginus tricuspis	Perodicticus potto	Tragelaphus spekii
V4			Cercocebus agilis	Galagoideus thomasi	Euoticus elegantulus	Colobus guereza	
V5			Hylochoerus meinertzhageni	Sciurocheirus gabonensis		Colobus satanas	
V6						Cercopithecus pogonias	

However, for species with lower encounter rates it is uncertain whether they were lost entirely in a particular area, or whether they were not observed. Overall, there were no species with high encounter rates missing entirely from either the FSC or the non-FSC concessions as can be seen in Extended Data Table 1.

Differentiating between real absence and low chances of detection is difficult, even in a study of this size and we do not want to draw too much attention to these differences as it would distract from the main results.

Also, the assertion that human population density is low around all sites, but it is not the same in all areas. I think some understanding of how human numbers (actually, a good surrogate of hunter density) and anthropogenic factors correlate with the results of the study is important.

We agree with the reviewer that human population density is a good surrogate for hunter density, and have included this information in the form of distance to roads and towns. This is, to our knowledge, the best indication of human population size we can attain. Following our additional

analysis, the impact of these covariates has now been discussed in the answer to the reviewers comment on lines 153-163.

Line 167: Better if the sentence states the importance of large mammals in the functioning of tropical forests because they are seed dispersers etc.

We thank the reviewer for the comment and have restructured the sentence to connect the importance of large mammals better to ecosystem functioning.

Lines 219 - 223:

“Conservation of large mammals through FSC certification brings wider benefits to forests, since the affected mammals play a pivotal role in delicate ecological processes such as seed dispersal, seed predation, browsing, trampling, plant competition, nutrient cycling and predator-prey interaction⁴⁰.”

Line 171: This sentence could be better described – Forest carbon storage is greater where there are more intact large mammal assemblages, because they allow large trees to flourish etc. as shown in Peres et al. (2016) Proc. Natl. Acad. Sci. 113, 892–897.

Again, we thank the reviewer for the comment and have restructured the sentence as suggested to make it more explicit how large mammals affect carbon storage.

Lines 223 - 227:

“Moreover, forest carbon storage is larger when large mammal assemblages are more intact, because the ecological processes they are part of (such as seed dispersal) often benefit large, high wood density trees⁴¹⁻⁴⁶, and the benefits of their conservation may far outweigh the cost⁴⁷.”

Lines 171-175: The assertion in the sentence “In addition, by being more biodiverse....” is unknown yet. There are many conditions and variables that influence zoonotic spread. It would be interesting to find out whether disease is less rife around better protected areas.

We agree that the assertion is at best weak and have removed the link between biodiversity and disease transmission accordingly.

Lines 227 - 229:

“In addition, by providing less wild meat for the markets, FSC-certified concessions or similar stringent schemes may also reduce the chance of zoonotic disease transmission⁴⁸.”

Line 185-186: The sentence “the differences between FSC-certified and non-FSC concessions that may drive wildlife abundance, were non-significant” is not substantiated in the paper since there are no data on the characteristics of each concession and the differences between them. This should be included as indicated above.

We thank the reviewer for the comment. The sentence refers to Fig. S1, which shows that geographic variation between FSC and non-FSC concessions is not significant. Despite this, we have now also included these covariates in our statistical model as discussed in the answer to the reviewers comment on lines 153-163.

Referee #5 (Remarks to the Author):

The manuscript by Zwerts et al. assesses the effect of FSC certification on mammal abundance, diversity and biomass. The authors do this by collecting camera trap data in FSC certified and non-

certified concessions in the Congo Basin. The results show that mammal encounter rate varies with certification status, with large mammals, critically endangered species and certain taxonomic groups being more often encountered in certified concessions than in non-certified ones. These results have large implications as they convincingly showed that timber extraction does not lead to overhunting when the right management practices (in this case following the FSC regulations) are applied, supporting the idea that forest concessions can play a pivotal role for mammal conservation.

The strong points of the manuscript are that 1) the questions being addressed are extremely relevant as the role of (certified) forest concessions/ forest management for biodiversity conservation is a contested issue in the conservation literature, 2) it provides empirical data that allows assessing if FSC certification reduces hunting pressure or not, 3) the study has a robust experimental design that includes multiple sites (14 concessions) and allows controlling for differences among sites (by using a paired site approach), 4) it is large-scale study, focussing a broad range of mammals, and using a large number of observations.

The points that can be improved on this manuscript are the following: 1) conservation implications of the results should be expanded (given the abovementioned strong points), 2) more-in-depth discussion on the practices needed to move towards a well-managed company with low negative impact on wildlife, 3) consideration of using a different statistical analysis, 4) the effect of time since logging is not explored in the analysis.

MAJOR COMMENTS

1. Conservation implications should be expanded given the importance of your findings for biodiversity conservation in Central Africa or elsewhere in the tropics. Several authors have claimed (without much evidence) that logging cannot be seen as a conservation tool because it opens the area to hunters, resulting in an increase in hunting pressure and further defaunation of the area. Your work shows very clearly (and with evidence!) that certified concessions do better than non-certified concessions in terms of mammal abundance, probably due to all the regulations and requirements that need to be followed by certified companies (e.g., closing road, patrols). Therefore, I think that you should expand the section on conservation implications of your discussion with the following aspects:

- Provide a larger view on the implications of your results than you do at this moment (lines 198-210). How do your results compare with other studies working on protected areas? And how do your results compare with studies done in areas that are not being managed for protection or for timber production? In other words, what is the contribution for biodiversity conservation for (non) certified companies when you compared them to protected areas and to forests without a clear manager? I can imagine that this is difficult to do as you do not have data on this, but these are issues and questions that have been in the literature without much empirical support, and I think that you have at least part of the data to explore these issues (with caution, obviously).

We thank the reviewer for the comment and have provided a more nuanced comparison between our results and study results in nearby protected areas. We discuss the implications of this comparison in relation to studies which emphasize the risk of logging as a conservation tool.

Lines 283 - 299:

“Of Central African tropical forests, 21% is designated for protection, but only 15% of the species ranges of central chimpanzees and the western lowland gorilla, lie in protected areas⁵⁵. Over half of these species ranges, and a large part of other mammals such as forest elephants, lie in logging concessions^{36,56}. Protected areas are essential for conservation, but sometimes lack the resources for effective control of illegal hunting^{57,58}. Forestry companies often do have the means to protect

forests and have an economic incentive to do so. We did not compare wildlife in protected areas with forestry concessions, but the encounter rates we observed of large mammals in FSC-certified concessions were comparable to those in recently monitored protected areas in the same region⁵⁹⁻⁶¹. Moreover, the relatively high encounter rates of large mammals (which are the first species to disappear as a result of hunting and poaching) in FSC-certified concessions compared to non-FSC concessions, suggest that wildlife communities in FSC-certified concessions resemble those in protected areas. The ratio of large versus small forest antelopes in the FSC-certified concessions is furthermore comparable to such ratios in a protected area in the region with almost no hunting, while those in non-FSC sites are vastly lower⁶². Our results challenge the notion that logging cannot be seen as a conservation tool^{23,63}, and we show that, if logged forests are properly managed, they can provide an important contribution to biodiversity conservation.”

- The potential role of well-managed concessions for landscape connectivity among protected areas. You mention this very briefly (line 207) but given the large distribution ranges of some of the species, I think that it is worthwhile to expand this part, and to discuss the role of managed forests in that regard.

We agree with the reviewer and have included a more thorough discussion about the potential role of well-managed forests in relation to landscape connectivity.

Lines 309 - 321:

“Most terrestrial protected areas are isolated⁶⁵, and increasing human modification of landscapes is limiting the ability for wildlife to move^{66,67}. Well-managed logging concessions can contribute to Sustainable Development Goals 12 (Sustainable Consumption and Production) and 15 (Life on Land) by performing a strategic function in preserving habitats and landscape connectivity while allowing for responsible economic activity⁶⁸. Concessions in our study region are large, often larger than 200,000 hectares, and together with protected areas they can have a substantial contribution to wildlife conservation. Governments in forest-rich countries may enhance the effectiveness of conservation policies by requiring FSC certification in strategic locations, such as buffer zones around protected areas to reduce the edge to area ratio of the conservation landscape⁶⁹. Non-FSC companies may also contribute to conservation, as they vary along a gradient of environmental and social responsibility⁷⁰. This was however not the focus of our study.”

- Put the size of the concessions into perspective as the concessions in your region are large and many readers will not be aware of their size. By doing this it becomes clear that concessions occupy large areas, and that together with protected areas can have a large contribution on wildlife conservation.

We thank the reviewer for the comment and have now emphasised the concessions sizes and potential conservation impact in our region:

Lines 314 - 316:

“Concessions in our study region are large, often larger than 200,000 hectares, and together with protected areas they can have a substantial contribution to wildlife conservation.”

2. More-in-depth discussion on the practices needed to move a non-certified company towards a well-managed company. You certainly discussed some of these aspects in the section “The necessity to upscale certification”, but I would like to see more specific recommendations in that regard. What are the basic practices listed among the Principal, Criteria and Indicators would you advice to start

with? I miss that kind of recommendations, which are important as such recommendation can be included in national codes of practices and forest laws.

We thank the reviewer for the comment. We acknowledge that it is indeed tempting to make specific recommendations about measures but believe this is beyond the scope of this study, which is detailed in lines 231 - 241:

“The FSC approach is holistic and controlling hunting is likely the most important factor for the reduction of environmental impacts in logged tropical forests⁷. We found more hunting signs in non-FSC concessions, which supports the interpretation that FSC effectively reduces hunting pressure, although we recognize that counting hunting signs are a relatively weak measure for the quantification of hunting pressure⁴⁹. Hunting has long been known to be the most important driver of forest fauna decline in Central African logged forests^{8,50} and the same phenomenon has recently been shown in Asia⁷. Nonetheless, other factors such as retaining high conservation value areas and reduced impact logging practices are likely to contribute to the observed effects as well⁵¹. Yet, our data do not allow for causal inference of the association of any of the specific measures implemented by FSC companies with the observed effects as that would require setting up more detailed measure-based experiments.”

3. Analysis was done using Wilcoxon signed-rank tests which is a very conservative test. I agree with the analysis done by the authors, but it is also a rather conservative analysis. I think that the authors could have done a mixed generalized linear model, using the data per camera trap. The model would be the following: camera trap nested within concession, paired concession as random factor, the fixed factor certification status (yes/no), and the different variables they measured. I think that in this way the authors could explore more the effect of certification status because their sample size would be much larger (28-36 camera x 14 concession) than the 14 data points they have at this moment. I think that the results would not change in direction but it is possible that stronger effects of certification would be found (as the test currently used is very conservative). Then the authors could run a similar analysis, with all other potential drivers of animal abundance (distance to settlements, roads, rivers, protected areas, elevation, time since logging, etc.) as fixed factors. In that way, they could assess how the different factors explain the response variables assessed.

We thank the reviewer for the comment and for the helpful suggestions. We have now used linear mixed models, which yielded higher statistical power and allowed for modelling the influence of the covariates. To account for potential non-linear effects along the covariate gradients we tested quadratic terms for all covariates which did not result in improved models.

The analytical approach is described in:

Lines 433 - 446:

“To assess whether encounter rates varied between FSC and non-FSC concessions, we quantified the means of the paired concessions using linear mixed-effects models with concessions, concession pairs and cameras as random effects. We allowed the means of concession pairs to vary between weight class, taxonomic group and IUCN Red List category if supported by model selection. We tested whether potential drivers of wildlife abundance (Figs. S1, S2, S3 & S4) variables were important using a model-selection approach based on minimization of Bayesian Information Criterion (BIC) values. We found that the inclusion of covariates did not substantially improve the model for weight classes and IUCN categories. For all mammals pooled together and for taxonomic groups however, the inclusion of elevation, which was correlated to distance to roads and rivers, slightly improved the model. There was however very little variation between encounter rates and elevation and all camera trap locations were non-mountainous lowland rainforests, so whatever

variation is linked to elevation is therefore assumed to be of very small importance (Fig. S6). Quadratic covariate terms did not result in better models. Post hoc comparisons were multivariate t adjusted.”

And discussed in:

Lines 246 - 249:

“While we believe that these covariates are important drivers of wildlife abundance⁵², including these covariates did not greatly improve the model, which underscores that camera grid locations were sufficiently similar in terms of these confounding influences.”

As the new modelling approach resulted in higher statistical power, Extended data figure 2 became redundant and is removed as all subgroups that were represented in this figure now have significant differences between the FSC and non-FSC groups.

4. Effect of time since logging is not tested even though we now that forest structure changes over time, and that these changes may vary between certified and non-certified companies (as damage is larger in the later). The results of such test should be included in Extended Data Table 6.

We thank the reviewer for the comment and we agree that forest structure changes over time since logging. Unfortunately though, it was practically impossible to collect reliable data about the time since logging for our sampling sites. Most of the logging companies roughly knew in which years the sampled areas were logged, but as logging operations were frequently delayed or changed, this data was unreliable. We did chose our sampling locations in sites that were logged 2-10 years before our study, although some camera grids overlapped older logging blocks.

This is mentioned in Lines 357 - 361:

“Camera trap grid locations within each pair of concessions were chosen based on similarity between potential drivers of animal abundance, including distance to settlements, roads, rivers, protected areas, elevation (Figs. S1; Table S5) and time since logging (2-10 years before our study), although some camera grids overlapped older logging blocks.”

Moreover, the effect of altered forest structure will be strongest immediately after logging, and as we did not include sites that were logged less than 2 years ago, we also believe that the impact of this on mobile mammals will not have a severe impact on our data.

We also discuss this issue in lines 249 - 253:

“Precise logging intensity and logging history data per camera were not available for most concessions because the companies’ planning schemes and actual exploitation of cutting blocks often did not match. Slight differences in logging history are not expected to have a large effect on the data, because wildlife is mobile and returns quickly to areas that have been exploited⁵³.”

MINOR COMMENTS

- Line 37-38: is one of the measures taken by certified companies the provision of alternative sources of meat? In the certified companies I have worked, the company would offer cow meat to workers (instead of a gun and bullets) to reduce the need to hunt. Is that the case also in your area? If so, please mention it.

We thank the reviewer for the remark as the certified companies certainly provide alternative sources of meat and we have now mentioned this in:

Lines 55 - 58:

“FSC certification may diminish these negative impacts because, among other measures, companies have to reduce accessibility to concessions by closing off old logging roads, prohibit wild meat transport or hunting materials, provide alternative sources of proteins and carry out surveillance by rangers.”

-Line 49-51: please revise the sentence. Negative effects of timber extraction ON WHAT?

We thank the reviewer for the observation, and have specified the sentence better.

Lines 79 - 81:

“Here we used camera traps to assess whether FSC certification can mitigate the negative effects of timber extraction on wildlife by studying the encounter rate of a broad range of mammal species across multiple sites.”

-Line 49-57: please revise the text to clarify better the situation of the concession in this region. The text now gives the idea that forest are intact and that the only disturbance is logging but then you say that hunting is pervasive. So logging is not the only disturbance, is it?

We have adapted the sentence to include hunting.

Lines 84 – 86:

“WEA is particularly suitable for these analyses, as its forests are reasonably intact and therefore its logging concessions are mostly devoid of influences other than the effects of logging and hunting²²⁻²⁴.”

-Line 71-72: provide more information about the study design in the legend of the figure, such as number camera per site, distance between camera. I think that this is important because the methods is provided much later.

We have added information about the study design in the legend figure:

Lines 103 - 105:

“Fig. 1. Locations of the 14 study sites in Gabon and the Republic of Congo. Between 28 to 36 cameras were deployed in each concession in systematic, one-kilometre spaced grids. Numbers and lines indicate the pairs of FSC-certified and non-FSC concessions.”

-Line: 198: strictly speaking the term “conservation” refers to the protection and sustainable use of a given area. So does the 21.1% you refer to include also concessions or is the value restricted to protected areas? If the latter is the case, then I would be explicit about it and talk about protection rather than conservation.

We thank the reviewer for the observation and have explicitly included that protection status of the 21.1% that we refer to.

Lines 283 - 285:

“Of Central African tropical forests, 21% is designated for protection, but only 15% of the species ranges of central chimpanzees and the western lowland gorilla, lie in protected areas⁵⁵”

-Line 198-200: I struggle with this sentence because it goes from % of conservation area to % of species distribution area found in concessions.

We understand the confusion and have adapted the sentence accordingly:

Lines 283 - 286:

“Of Central African tropical forests, 21% is designated for protection, but only 15% of the species ranges of central chimpanzees and the western lowland gorilla, lie in protected areas⁵⁵. Over half of these species ranges, and a large part of other mammals such as forest elephants, lie in logging concessions^{36,56}.”

-Line 200-201: this sentence is kind of lost in the arguments being presented in this paragraph. I think that you can elaborate much more than what you have done now. Please my first major comment above.

Please see our response your major comment.

-Line 211: please rephrase as “FSC effectively controls” is not correct. FSC does not control itself, the rules/requirements/regulations of “FSC certification lead to a more effective control”.

We have adapted sentence based on the reviewers’ comment:

Lines 322 - 324:

“Our findings indicate that the regulations of FSC certification lead to more effective control of widespread and unsustainable hunting and poaching that is facilitated by the increased access to forests engendered by timber extraction.”

-Line 255: Indicate how many cameras were then put over time in the 14 concessions, and which area was covered by the cameras per concession.

We have now included the total number of cameras and area covered and made reference to Table S4.

Lines 368 - 369:

“Between 28 to 36 cameras were deployed in each concession, totalling up to 474 camera traps, distributed over 474 km² (Table S4).”

-Line 271-275: what proportion of pictures were then not used because the species could not be identified or the rate observations?

We thank the reviewer for the comment and wish to point out that the encounter rates of genera that were not identified to the species level (*Cercopithecus* indet, *Cephalophus* indet) are displayed in Extended Data Table 1.

Reviewer Reports on the First Revision:

Referees' comments:

Referee #2 (Remarks to the Author):

GENERAL

The manuscript is improved and very close to publication.

I am slightly worried that the FSC affiliations of two of the authors might be leading them to promote FSC logging in their Discussion. Certainly, they have shown that FSC is better than non-FSC, at least for mammals – I am not questioning their fascinating data. But there was no attempt to compare FSC to non-logged areas. The best circumstances for your average large mammal in West/Central Africa is no logging at all combined with regular enforcement. So FSC is very much a second best compared to full protection but the Discussion has a flavour of let's promote good logging as a way to look after these forests. This worries me and it needs addressing explicitly.

SPECIFIC

28 what are the challenges?

30 more wildlife – abundance or species?

75 why dropped one – what was special about it?

Figures 3, 4 and 5 are very interesting

252-253 very strong statement – you have only monitored mammals, and only in West Africa. I would qualify or omit this.

278-279 my suspicion is that it not the certification so much as the control of compliance and regular enforcement and you did not explicitly test for these issues.

349 why is an interaction needed – shouldn't you just test for more sightings with time in the non-fsc areas?

Tim Caro

Referee #3 (Remarks to the Author):

I have made some small edits in the new text (in the rebuttal letter). I am generally very happy with the changes the authors have made to the reviewers' comments and believe the paper is much stronger. I would urge the authors to go over the manuscript and ensure that all statements are clear, especially those I have highlighted.

Referee #4 (Remarks to the Author):

I have read the revised manuscript by Zwerts et al. assessing the effect of FSC certification on mammal abundance, diversity and biomass. I have also read the rebuttal letter provided by the authors.

My major comments were: 1) conservation implications of the results should be expanded (given the abovementioned strong points), 2) more-in-depth discussion on the practices needed to move towards a well-managed company with low negative impact on wildlife, 3) consideration of using a different statistical analysis, 4) the effect of time since logging is not explored in the analysis.

I believe that the authors have addressed my comments and the comments provided by three

other reviewers adequately. My comments at this moment are mostly editorial, some of them included in the PDF itself (see attached document), and others included below. My main comments at this stage is that the authors need to be consistent with terminology used in the text and in the literature (see suggestions below) to increase clarity. The text may also benefit from a English proof reading.

Minor comments:

Line 112; what do you mean with marked differences? The results were not significant, so I think that you should say that you did not find statistically significant differences.

Line 115-120. I think that you can improve the legend of this figure. For example in line 118 you can indicate what is the fixed factor and what are the random factors as the reader has not really read the methods yet. Indicate that results of A come from X test and of B from another test.

Apply these changes in other legends if needed.

Line 179-181: move text to line 110 as you report here the effect of concession type on the total encounter rate.

Line 187 and elsewhere in the discussion: the term FSC-certified forestry is not commonly used in the literature I am familiar with. I would say forest management certified under the FSC scheme, and in short "FSC-certified forest management". I also find strange that you are changing terminology in discussion after using other terms elsewhere in the document.

Line 192-191: revise statement "because the latter recover more slowly from population losses and may be targeted more by 192 hunters" as you are only talking about species >10 kg.

Line 214-215: this sentence should be moved earlier in the text about pangolins so the reader understands better why you focus on this species.

Line 215-216: move this paragraph to the results section as you do not really discuss them here. Did you report the results on diversity in the result section?

Line 231: the statement "The FSC approach is holistic" is rather vague for a reader who is not familiar with the scheme. I think that the authors need to indicate why is FSC holistic, or start the paragraph with the second part of the sentence "Controlling hunting....".

Lines 288-289: is the statement "Forestry companies often do have the means to protect forests and have an economic incentive to do so" valid for all companies or only the ones that have concessions certified by FSC?

Lines 288-289: not sure if forestry companies is the right term here. Do you mean companies with forest concession, do not you? Also the term forestry concession should read forest concession, or timber concession. Please revise the whole document.

Line 306: ...are also applied in other FSC-certified tropical forestry systems – change the previous statement to: ... are also applied in other FSC-certified areas as these aspects are part of the FSC principles, criteria and indicators (REF TO TABLE).

Line 309-321: consider writing this part as a separate paragraph to make a stronger case. For this you would need to reshuffle this section slightly to bring the message upfront (now it is towards the end of the section).

Line 322: change regulations for requirements. I do not think that regulation is the right terms as this is a voluntary system.

Line 676-681: why is this figure needed when it has been included as well as Fig. 3?

Line 731-732: I think that the legend needs to include more information on the test done (I guess here you are reporting the Linear Mixed Model results, are you not?), including the random factors used in each case. Please also explain abbreviations. I do not think that you need to include all the STA variables (Sumsq and Mean sq are repetitive information). Please explain abbreviations used.

Line 825: Table S6 is missing

Referee #5 (Remarks to the Author):

This is a very nice study. I was asked, in particular, to review the camera trap methodology and analyses. I found them to be quite strong and only have a few comments below. I think their study design is quite robust and their handling of the camera trap data was appropriate.

Counter to one of the other reviewers, I'm quite ok considering detection rate as a measure of relative abundance. While animal activity can also affect detection rate, many studies have found a strong relationship to animal density (e.g. Parsons 2017) , and I see no reason animals would move so much more when hunted (I would expect they move less, actually, but not sure if there are papers on that). Furthermore, I think they should point out that this detection rate measure is directly relevant to the ecological impact of these species, regardless of whether its reflecting density or movement.

Parsons, A. W., Forrester, T., McShea, W. J., Baker-Whatton, M. C., Millsaugh, J. J., & Kays, R. (2017). Do occupancy or detection rates from camera traps reflect deer density? *Journal of Mammalogy*, 98, 1547–1557. <https://doi.org/10.1093/jmammal/gyx128>

276-281 Point out that increased activity of these species would also result in increased ecological impacts.

277 – what is 'activity'? I think its actually movement per day, might clarify that.

279 – might want to rename your measure of biomass 'relative biomass' or 'biomass index' to clarify its not a 'true biomass'.

394 – was this group size, number of animals, used in the analysis? If not, don't need to mention it. If so, please clarify that it was not simple detection rate but weighted by group size.

406 – I don't think there are really any other factors that are important other than movement rate.

407 – I don't see how 'diurnal activity rates' would matter, movement per day, yes, but not rather the activity is day or night.

409 - please clarify if these observations were weighted by group size or not.

669 – important that the raw detection data are made available.

Referee #6 (Remarks to the Author):

Review for manuscript 2022-10-16740A "FSC-certified forestry benefits large mammals compared to non-FSC"

This paper assesses the effects of FSC certification on wildlife in Western African tropical forests using a large dataset of camera trap data. The study found that FSC certified forests contain more large and high conservation priority species, while non-certified forests contain proportionally more small species than certified concessions. The study concludes that FSC-certified forestry is less damaging to the mammal community than non-certified forestry. The outstanding features of this study are well summarised by the current set of reviews, and I have restricted my review to a consideration of the responses to the changes suggested by reviewer 1, primarily the ability to separate the influence of hunting from other aspects of FSC certification and the statistical methods, i.e. the addition of the linear mixed model to the analysis.

Reviewer 1 raised several major issues which have been addressed in the authors' response. I deal with each of these in turn before providing specific minor comments. For the most part I was satisfied that the authors have addressed the reviewer's concerns. However, although the broad

statistical methods selected are appropriate, I had some difficulty verifying the linear mixed modelling approach due to a lack of detail about the model structure.

Line numbers refer to the tracked changes copy of the manuscript.

Issue 1: Separation of the influence of hunting from other aspects of certification. I agree with Reviewer 1 that the influence of hunting is not explicitly studied in this manuscript and that the authors cannot attribute the observed differences in encounter rate to hunting alone. Despite this, the authors have circumstantial evidence of increased hunting pressure in the non-certified concessions (via measuring hunting sign) and have amassed a convincing body of literature as evidence in their rebuttal to suggest that a reduction in hunting pressure is the likely mechanism for the increases in large mammal encounter rate (at least in tropical West African forests). For the most part the authors have done an acceptable job of referring to the link between increased mammal encounters and hunting as a likely effect rather than causal. There is one place in the manuscript where the link to hunting is overstated and should be reworded (lines 322-324, see specific comments).

Issue 2: Generality of the study to the global context. This has been satisfactorily addressed by the authors, who have also done a good job of addressing the caveats associated with extending the findings to other contexts.

Issue 3: Reference to encounter rates as a proxy for abundance: This has been well addressed by the authors.

Issue 4: Choice of analytical model and the need to better account for covariates: The use of the linear mixed model instead of the two-sided Wilcoxon signed rank tests allows the authors to model the effects of correlated groupings (here concessions, concession pairs and cameras) on encounter rate. Reviewer 1 suggested a hierarchical model structure, for which mixed models are an appropriate choice, however I found it difficult to assess the actual model structure that was used in this study as the text only notes the random effects that were used without detailing the structure of the random effects (presumably cameras were nested within concessions within concession pairs, in a multi-level random effect structure?). The text also notes that several models were tested, then sorted using BIC—again this is an appropriate choice, but it would be useful to see the model structures that were tested to help the reviewers assess the methods. I note that the code will be made available for interested readers upon publication, however it would be useful if the authors provided a list of the models that were tested as part of the supporting information.

The reviewer's concerns about interactions between covariates appear to have been addressed by the authors' consideration of the quadratic covariate terms.

Issue 5: Accuracy of the hunting sign method: I am not well qualified to answer this concern as monitoring methods to detect hunting presence are outside my expertise, however the author's acknowledgement of the weaknesses of this approach in the manuscript appears to address this issue from my layman's perspective.

Specific comments:

Abstract, line 24: "Over a quarter of the tropical forests is exploited for timber". This is grammatically incorrect—rephrase (possibly "Over a quarter of the world's tropical forests are exploited for timber"?).

Line 76 "Hunting not necessarily completely extirpates wildlife species...": Rephrase—e.g. "hunting does not necessarily completely extirpate..."

Line 131: "Mammal encounter rates of all IUCN Red List categories were higher"—the p-values in Figure 4 and Extended Data Table 3 suggest that this is not true for Endangered and Vulnerable species. Could the authors explain or rephrase this sentence?

Line 176: "We detected no difference in overall species diversity". In extended table 4, there are

several species that are detected only in either the FSC or non-FSC certified areas (for example there are at least 6 species of primates that were not detected in the FSC-certified areas). Was this an expected outcome, and are there systematic differences in faunal composition that are worth noting here?

Line 185 "the loss of large wildlife" (and elsewhere). There are several places where the authors use "wildlife" (e.g. lines 252, 254, 305, 336 etc) or "animals" (line 359, 690), which may be taken out of context since the study only examines mammal species. It would be more precise to use "mammals" everywhere.

Line 231: "The FSC approach is holistic and controlling hunting is likely the most important factor for the reduction of environmental impacts in logged tropical forests." The phrase "The FSC approach is holistic" may be confusing to readers who don't have the context of the review rebuttal. Consider rephrasing to "FSC certification must satisfy several criteria to support holistic forest management. In logged tropical forests, controlling hunting is likely the most important factor..." (or similar wording—edit as required)

Lines 253: "the geographic variation... was non-significant". Is this a statement of statistical significance or a general statement meaning "not important"?

Line 278: "Species home ranges and movement patterns can change in response to disturbance, which can affect encounter rates". This apparently contradicts with the argument in line 251, i.e. "Slight differences in logging history are not expected to have a large effect on the data because wildlife is mobile and returns quickly to areas that have been exploited". Am I misinterpreting the latter statement here?

Lines 322-324: The findings don't indicate that the FSC regulations lead to more effective control of unsustainable hunting. As noted at line 239, the data in this study don't allow for causal inference about the specific measures implemented as part of the FSC regulations, so the link to hunting is a hypothesis that is not tested in this study. This sentence should be reworded accordingly.

Note that Table S6 was missing from my review package, so I have not reviewed the material beyond Table S5.

Author Rebuttals to First Revision:

Manuscript tracking number 2022-10-16740A

Response to referees

Line numbers refer to the tracked changes copy of the manuscript.

Referees' comments:

Referee #2 (Remarks to the Author):

GENERAL

The manuscript is improved and very close to publication.

We thank the reviewer for the comment and are happy to hear the previous edits gained approval.

I am slightly worried that the FSC affiliations of two of the authors might be leading them to promote FSC logging in their Discussion. Certainly, they have shown that FSC is better than non-FSC, at least for mammals – I am not questioning their fascinating data. But there was no attempt to compare FSC to non-logged areas. The best circumstances for your average large mammal in West/Central Africa is no logging at all combined with regular enforcement. So FSC is very much a second best compared to full protection but the Discussion has a flavour of let's promote good logging as a way to look after these forests. This worries me and it needs addressing explicitly.

We comprehend that the two authors individual FSC memberships can be interpreted as slightly worrying, but we would like to point out that for these authors, there are no personal stakes or benefits. It is also important to note that if the data would have shown neutral or the opposite results, we would have also published our results.

We agree with the reviewer that the first priority for species protection is no logging combined with effective law enforcement, and have rephrased the sentence about logging as a conservation tool in lines 290 - 296:

“The ratio of large versus small forest antelopes in the FSC-certified concessions is furthermore comparable to such ratios in a protected area in the region with almost no hunting, while those in non-FSC sites are far lower⁵⁷. Although the first priority for species protection should be to maintain unlogged forests where there is effective law enforcement, our results challenge the notion that, at least for large-bodied mammals in WEA, logging is always disastrous^{58,59}”

We also removed the concluding remark about the efficacy of FSC certification as a conservation tool in lines 336 – 339:

“We present a clear, evidence-based message about the positive impact of FSC certification. We show that medium to large-sized mammals – that play vital functions in forests – are more abundant in FSC-certified concessions than in non-FSC concessions.”

We did not compare FSC to non-logged areas due to the large variety in effectiveness of enforcement of protected and non-logged forests in WEA, which was likely to have given a skewed representation of wildlife in non-logged forests, rendering a regional comparison between FSC and non-logged forests less useful.

We respond to the reviewers' concern in more detail below in response to the comment about lines 252-253.

SPECIFIC

28 what are the challenges?

We have now specified the most important research design challenge for land-use impact studies, namely constraints in relation to sample size. We have only mentioned the most important challenge in view of the word limit of the abstract and considering that more specific examples are present in the introduction.

Lines 28 - 31: "Forest management certification schemes such as the Forest Stewardship Council (FSC) are expected to mitigate impacts on biodiversity but to date very little is known about the effectiveness of FSC certification due to research design challenges, predominantly due to limited sample sizes^{2,3}."

30 more wildlife – abundance or species?

We have now specified that this statement concerns mammal abundances.

Lines 32 - 34: "We observed higher mammal encounter rates in FSC-certified than in non-FSC logging concessions."

75 why dropped one – what was special about it?

We have now specified that the reason that we did not include all companies in the region was because the respective company did not allow access to their concessions.

Lines 80 - 88: "We compared small to large-sized mammal observations across seven paired FSC-certified and non-FSC concessions in Gabon and the Republic of Congo (Fig. 1). Gabon and the Republic of Congo lie within Western Equatorial Africa (WEA) and we included all companies that were FSC-certified between 2018 and 2021 in this region, with the exception of one which refused to allow access. WEA is particularly suitable for these analyses, as its forests are reasonably intact and logging concessions are embedded in a matrix of contiguous forest, which are therefore mostly devoid of influences other than the effects of logging and hunting^{20,21}."

Figures 3, 4 and 5 are very interesting

We thank the reviewer for the comment and certainly agree.

252-253 very strong statement – you have only monitored mammals, and only in West Africa. I would qualify or omit this.

We have explicitly mentioned that non-logged forests with effective enforcement should be prioritized over well-managed logged forests. We also specified that our results specifically address mammals in Western Equatorial Africa:

Lines 293 - 296:

"Although the first priority for species protection should be to maintain unlogged forests where there is effective law enforcement, our results challenge the notion that, at least for large-bodied mammals in WEA, logging is always disastrous^{58,59}."

We also would like to point out the consideration discussed prior to the statement in lines 288 - 291, which serves as a build-up as explanation to the statement about logging as a conservation tool:

Lines 281 - 293: “Protected areas are essential for conservation, but sometimes lack the resources for effective control of illegal hunting^{52,53}. Logging companies often do have the means to protect forests and have an economic incentive to do so. We did not compare mammal encounter rates in protected areas with the same metric in logging concessions ourselves, but the encounter rates we observed of large mammals’ in FSC-certified concessions were comparable to published data from recently monitored protected areas in the same region^{54–56}. Moreover, the relatively high encounter rates of the largest mammals (which are the first species to disappear as a result of hunting and poaching) in FSC-certified concessions compared to non-FSC concessions, suggest that mammal communities in FSC-certified concessions resemble those in protected areas. The ratio of large versus small forest antelopes in the FSC-certified concessions is furthermore comparable to such ratios in a protected area in the region with almost no hunting, while those in non-FSC sites are far lower⁵⁷.”

Moreover, we would also like to point out we have specified that extrapolation of our results to other regions should be done with caution.

Lines 304 - 307: “We infer this with caution as timber extraction volumes, concession size and shape, presence of public roads, population density and other characteristics may differ between concessions and thereby affect the impacts of FSC-certified forest management⁶⁰.”

278-279 my suspicion is that it not the certification so much as the control of compliance and regular enforcement and you did not explicitly test for these issues.

We agree with the reviewer that control of compliance and regular enforcement are very important, but that these are, at least in West and Central Africa, not always effectively implemented by the authorities. FSC does ensure their implementation, which serves as an explanation for the results that we found. We have slightly adapted the sentence to include that FSC ensures all these factors contributing to successful protection of wildlife.

Lines 332 - 334: “We believe that a strict set of requirements, control of compliance and regular enforcement, all integrally connected and ensured within the FSC system, are crucial for successful environmental protection through forest certification.”

349 why is an interaction needed – shouldn’t you just test for more sightings with time in the non-fsc areas?

Because showing more sightings with time in the non-FSC areas would not provide a definite answer as there might also be more sightings with time in the FSC areas, which would not be tested without an interaction. Therefore, we test the interaction to be able to infer whether certification status differentially affects sightings over time.

Tim Caro

We thank the reviewer for his comments, it was a pleasure.

Referee #3 (Remarks to the Author):

I have made some small edits in the new text (in the rebuttal letter). I am generally very happy with the changes the authors have made to the reviewers' comments and believe the paper is much stronger. I would urge the authors to go over the manuscript and ensure that all statements are clear, especially those I have highlighted.

We thank the reviewer for edits made in the reviewers' comments document and we have implemented all suggestions.

Referee #4 (Remarks to the Author):

I have read the revised manuscript by Zwerts et al. assessing the effect of FSC certification on mammal abundance, diversity and biomass. I have also read the rebuttal letter provided by the authors.

My major comments were: 1) conservation implications of the results should be expanded (given the abovementioned strong points), 2) more-in-depth discussion on the practices needed to move towards a well-managed company with low negative impact on wildlife, 3) consideration of using a different statistical analysis, 4) the effect of time since logging is not explored in the analysis.

I believe that the authors have addressed my comments and the comments provided by three other reviewers adequately. My comments at this moment are mostly editorial, some of them included in the PDF itself (see attached document), and others included below. My main comments at this stage is that the authors need to be consistent with terminology used in the text and in the literature (see suggestions below) to increase clarity. The text may also benefit from a English proof reading.

We thank the reviewer for edits made in the PDF and we have implemented all suggestions except for two instances.

First, we did not implement the suggestion to add the word "currently" in line 83, because new companies have become FSC-certified in the study region since we have gathered the data. We have thus specified the sentence as follows:

Lines 81 – 85: "We compared small to large-sized mammal observations across seven paired FSC-certified and non-FSC concessions in Gabon and the Republic of Congo (Fig. 1). Gabon and the Republic of Congo lie within Western Equatorial Africa (WEA) and we included all companies that were FSC-certified between 2018 and 2021 in this region, with the exception of one which refused to allow access."

Second, we did not add "as shown in Figure 1" in the figure texts of figure 2, 3 and 4. But instead edited the text to shorten it and make the figure texts consistent throughout the document:

Lines 143 – 146: "Numbers represent paired FSC-certified (n=7) and non-FSC (n=7) concessions, red lines represent Linear Mixed Model predicted fixed effects (certification status) and grey lines represent random effects (concession pairs)."

We have applied these changes where relevant in other figure legends as well.

A native English speaker has now proofread the document.

Minor comments:

Line 112; what do you mean with marked differences? The results were not significant, so I think that you should say that you did not find statistically significant differences.

We thank the reviewer for the comment and improved the wording of the sentence to reflect that we report on the observed species numbers in FSC and non-FSC concessions:

Lines 121 – 124: “We observed comparable species diversity in the two concession types, as only a small number of species, all with very low encounter rates, were lacking completely in one or other of the concession types.”

Line 115-120. I think that you can improve the legend of this figure. For example in line 118 you can indicate what is the fixed factor and what are the random factors as the reader has not really read the methods yet. Indicate that results of A come from X test and of B from another test. Apply these changes in other legends if needed.

We thank the reviewer for the comment and improved the figure legend:

Lines 127 - 133: “Fig. 2. Mammal encounter rate and hunting signs in FSC-certified and non-FSC concessions. (A) Encounter rate of all observed mammals, and (B) proportion of camera locations with hunting signs. Numbers represent paired FSC-certified (n=7) and non-FSC (n=7) concessions. The red line in panel A represents the Linear Mixed Model predicted fixed effect (certification status) and grey lines represent random effects (concession pairs). Differences between hunting signs in panel B were analysed using a Wilcoxon signed-rank test. Boxplot whiskers reflect 1.5*IQR. *: p < 0.05.”

We have applied these changes where relevant in other figure legends as well.

Line 179-181: move text to line 110 as you report here the effect of concession type on the total encounter rate.

Thank you for this observation, we have moved the text.

Lines 115 - 124: “We detected 55 mammal species and found a positive effect of FSC-certification on overall mammal encounter rate (Fig. 2A). We also found fewer signs of hunting (Fig. 2B) in FSC-certified than in non-FSC concessions (Extended Data Table 2 & Supplementary Table 3). Estimated total faunal biomass derived from mammal encounter rates was 4.5 times higher in FSC-certified compared to non-FSC concessions. Larger species contributed more to the total biomass (Extended Data Fig. 1). We observed comparable species diversity in the two concession types, as only a small number of species, all with very low encounter rates, were lacking completely in one or other of the concession types.”

Line 187 and elsewhere in the discussion: the term FSC-certified forestry is not commonly used in the literature I am familiar with. I would say forest management certified under the FSC scheme, and in short “FSC-certified forest management”. I also find strange that you are changing terminology in discussion after using other terms elsewhere in the document.

We thank the reviewer for the observation and have adapted the text throughout the document, including the title.

Line 192-191: revise statement “because the latter recover more slowly from population losses and may be targeted more by 192 hunters” as you are only talking about species >10 kg.

We thank the reviewer for the observation and have revised the sentence accordingly.

Lines 181 - 183: “This effect was most pronounced for species larger than 10 kg, which was consistent for all FSC-non-FSC concession pairs, likely because these medium to large species recover more slowly from population losses, and may be targeted more by hunters^{25,26}.”

Line 214-215: this sentence should be moved earlier in the text about pangolins so the reader understands better why you focus on this species.

We agree with the reviewer and have adapted the text accordingly.

Lines 197 - 204: “In contrast, no difference was found in pangolin encounter rate, which are among the most trafficked mammals, between the two types of logging regimes³⁵. Two out of the three pangolin species present in WEA are relatively small and generally have higher reproduction rates than mammals in larger size classes. Moreover, all three pangolin species had low encounter rates in our study (Supplementary Table 3), likely because two pangolin species are semi-arboreal and are therefore not effectively captured by ground-based camera traps, which reduces our ability to draw strong conclusions about these species and warrants further research.”

Line 215-216: move this paragraph to the results section as you do not really discuss them here. Did you report the results on diversity in the result section?

We thank the reviewer for the comment but would like to point out that we reported the diversity results in lines 121 - 122:

“We observed comparable species diversity in the two concession types.”

Since this is mentioned in the results section, we see limited added value in moving the paragraph to the result section. Moreover, we prefer to retain this text in the discussion to explain why we did not find difference in species diversity.

Line 231: the statement “The FSC approach is holistic” is rather vague for a reader who is not familiar with the scheme. I think that the authors need to indicate why is FSC holistic, or start the paragraph with the second part of the sentence “Controlling hunting....”.

We agree with the observation and have adapted the text accordingly.

Lines 221 - 226: “The FSC takes a comprehensive and all-encompassing perspective when it comes to managing and promoting sustainable forest practices. This approach recognizes that forests are complex ecosystems with intricate interconnections between their various components, including flora, fauna, soil, water, and climate. In logged tropical forests, controlling hunting is likely the most important factor for the reduction of environmental impacts in logged tropical forests⁷.”

Lines 288-289: is the statement “Forestry companies often do have the means to protect forests and have an economic incentive to do so” valid for all companies or only the ones that have concessions certified by FSC?

We thank the reviewer for the comment and would like to make use of the opportunity for our motivation of this sentence. The cost of protecting biodiversity can be high for conservation agencies that have to establish a presence on the ground, but can be relatively limited for a company that has to manage and their concessions and property. Yet, we cannot assert with certainty whether this is indeed feasible for all forestry companies, hence the word ‘often’ is used in the respective sentence. We do however not have data to confidently make a distinction between the economics of FSC-certified and non-FSC forestry companies.

Lines 282 - 283: "Logging companies often do have the means to protect forests and have an economic incentive to do so."

Lines 288-289: not sure if forestry companies is the right term here. Do you mean companies with forest concession, do not you? Also the term forestry concession should read forest concession, or timber concession. Please revise the whole document.

We thank the reviewer for the comment and have changed 'forestry concession' to 'logging concession', and 'forestry company' to 'logging company', throughout the document to ensure consistent terminology.

Line 306: ...are also applied in other FSC-certified tropical forestry systems – change the previous statement to: ... are also applied in other FSC-certified areas as these aspects are part of the FSC principles, criteria and indicators (REF TO TABLE).

Done.

Lines 299 - 304: "The results of this study are likely to be applicable to other logged tropical forests where hunting, through increased accessibility, poses a risk to forest mammals. This is because wildlife protection measures and law enforcement are applied across all FSC-certified forests, as part of the FSC principles, criteria and indicators (Supplementary Tables 1 & 2)."

Line 309-321: consider writing this part as a separate paragraph to make a stronger case. For this you would need to reshuffle this section slightly to bring the message upfront (now it is towards the end of the section).

We thank the reviewer for the comment and agree that the topic of fragmentation deserves more attention. We have created a separate paragraph to discuss the issue and have adapted the text accordingly to emphasize the message.

Lines 308 - 319: "Most terrestrial protected areas are isolated⁶¹, and increasing human modification and fragmentation of landscapes is limiting the ability of mammals to move^{62,63}. Governments in forest-rich countries may enhance the effectiveness of conservation policies by requiring FSC certification in strategic locations, such as buffer zones around protected areas to reduce the edge to area ratio of the conservation landscape⁶⁴. Non-FSC companies may also contribute to conservation, as they vary along a gradient of environmental and social responsibility⁶⁵. This was however not the focus of our study. Concessions in our study region are large, often larger than 200,000 hectares, and together with protected areas they can have a substantial contribution to mammal conservation. Well-managed logging concessions can contribute to Sustainable Development Goals 12 (Sustainable Consumption and Production) and 15 (Life on Land) by performing a strategic function in preserving habitats and landscape connectivity while allowing for responsible economic activity⁶⁶."

Line 322: change regulations for requirements. I do not think that regulation is the right terms as this is a voluntary system.

We thank the reviewer for the sharp observation and have adapted the text accordingly.

Line 325: "Our findings indicate that the requirements of FSC certification ..."

We have replaced the word regulations in the context of FSC certification throughout the document, such as in lines 330 - 334:

“Sustainability of this practice is controlled by - among other requirements - controlling firearm permits, spatially assigning hunting zones, and monitoring wildlife offtake. We believe that a strict set of requirements, control of compliance and regular enforcement, all integrally connected and ensured within the FSC system, are crucial for successful environmental protection through forest certification.”

Line 676-681: why is this figure needed when it has been included as well as Fig. 3?

This figure was included to provide a relative overview of the encounter rates between the weight classes, but we agree that including the figure is somewhat redundant if it only serves this purpose. We have thus removed the figure from the manuscript.

Line 731-732: I think that the legend needs to include more information on the test done (I guess here you are reporting the Linear Mixed Model results, are you not?), including the random factors used in each case. Please also explain abbreviations. I do not think that you need to include all the STA variables (Sumsq and Mean sq are repetitive information). Please explain abbreviations used.

We revised the table by removing the Sum of Squares column and have written out all abbreviates in full. For each analysis we have also specified the random effects.

Line 825: Table S6 is missing

We apologize for the inconvenience, which may have arisen because the table is included as a separate excel file. We will make sure the table is attached to the current version of the manuscript.

Referee #5 (Remarks to the Author):

This is a very nice study. I was asked, in particular, to review the camera trap methodology and analyses. I found them to be quite strong and only have a few comments below. I think their study design is quite robust and their handling of the camera trap data was appropriate.

We thank the reviewer for the kind words.

Counter to one of the other reviewers, I'm quite ok considering detection rate as a measure of relative abundance. While animal activity can also affect detection rate, many studies have found a strong relationship to animal density (e.g. Parsons 2017), and I see no reason animals would move so much more when hunted (I would expect they move less, actually, but not sure if there are papers on that). Furthermore, I think they should point out that this detection rate measure is directly relevant to the ecological impact of these species, regardless of whether it's reflecting density or movement.

Parsons, A. W., Forrester, T., McShea, W. J., Baker-Whatton, M. C., Millspaugh, J. J., & Kays, R. (2017). Do occupancy or detection rates from camera traps reflect deer density? *Journal of Mammalogy*, 98, 1547–1557. <https://doi.org/10.1093/jmammal/gyx128>

276-281 Point out that increased activity of these species would also result in increased ecological impacts.

We thank the reviewer for the comment and we agree that it is unlikely that changes in activity solely make up the observed differences in encounter rates, given the consistency of the data in the three highest weight classes.

We have now specified this in lines 273 - 275: "It is however unlikely that changes in activity solely make up the observed differences in encounter rates, given the consistency of the data in the three heaviest weight classes."

We cannot, unfortunately, assert with certainty that increased activity results in increased ecological impacts, because we do not have the data to back this up. Neither can we address the impact of increasing activity only, as detection rates do not provide us with the ability to assess whether activity increases or decreases. But these are certainly interesting questions to address in future studies.

277 – what is 'activity'? I think its actually movement per day, might clarify that.

We thank the reviewer for the observation and we have clarified the sentence accordingly.

Lines 271 - 272: "... in abundance, activity – movement per day –, or both."

279 – might want to rename your measure of biomass 'relative biomass' or 'biomass index' to clarify its not a 'true biomass'.

We thank the reviewer for the suggestions and have added the word relative before biomass.

Line 275: "We also estimate relative biomass using encounter rates, ..."

394 – was this group size, number of animals, used in the analysis? If not, don't need to mention it. If so, please clarify that it was not simple detection rate but weighted by group size.

We thank the reviewer for the sharp observation and we can confirm that group size was used in the analysis. We have now specified this in lines 577 - 579:

"For each species for each concession, we calculated encounter rate, weighted by group size, as the number of observations divided by the sampling effort and we reported all findings using the metric "Observations / day"."

406 – I don't think there are really any other factors that are important other than movement rate.

We agree with the reviewer that this is highly unlikely, but because of our use of encounter rates instead of calculating true density, there is a chance that detection probability is influenced by other factors which is why we chose to retain this sentence.

407 – I don't see how 'diurnal activity rates' would matter, movement per day, yes, but not rather the activity is day or night.

We thank the reviewer for the comment and have adapted the sentence accordingly.

Lines 574 -576:

"We recognize however that other factors may exist that may have influenced detection probability, such as movement rates, which may be affected by hunting."

409 - please clarify if these observations were weighted by group size or not.

We have now specified this in lines 577 - 579:

“For each species for each concession, we calculated encounter rate, weighted by group size, as the number of observations divided by the sampling effort and we reported all findings using the metric “Observations / day”.”

669 – important that the raw detection data are made available.

We agree. All data and R code will be uploaded to Zenodo depository before publication.

Referee #6 (Remarks to the Author):

Review for manuscript 2022-10-16740A “FSC-certified forestry benefits large mammals compared to non-FSC”

This paper assesses the effects of FSC certification on wildlife in Western African tropical forests using a large dataset of camera trap data. The study found that FSC certified forests contain more large and high conservation priority species, while non-certified forests contain proportionally more small species than certified concessions. The study concludes that FSC-certified forestry is less damaging to the mammal community than non-certified forestry. The outstanding features of this study are well summarised by the current set of reviews, and I have restricted my review to a consideration of the responses to the changes suggested by reviewer 1, primarily the ability to separate the influence of hunting from other aspects of FSC certification and the statistical methods, i.e. the addition of the linear mixed model to the analysis.

Reviewer 1 raised several major issues which have been addressed in the authors’ response. I deal with each of these in turn before providing specific minor comments. For the most part I was satisfied that the authors have addressed the reviewer’s concerns. However, although the broad statistical methods selected are appropriate, I had some difficulty verifying the linear mixed modelling approach due to a lack of detail about the model structure.

Line numbers refer to the tracked changes copy of the manuscript.

Issue 1: Separation of the influence of hunting from other aspects of certification. I agree with Reviewer 1 that the influence of hunting is not explicitly studied in this manuscript and that the authors cannot attribute the observed differences in encounter rate to hunting alone. Despite this, the authors have circumstantial evidence of increased hunting pressure in the non-certified concessions (via measuring hunting sign) and have amassed a convincing body of literature as evidence in their rebuttal to suggest that a reduction in hunting pressure is the likely mechanism for the increases in large mammal encounter rate (at least in tropical West African forests). For the most part the authors have done an acceptable job of referring to the link between increased mammal encounters and hunting as a likely effect rather than causal. There is one place in the manuscript where the link to hunting is overstated and should be reworded (lines 322-324, see specific comments).

Issue 2: Generality of the study to the global context. This has been satisfactorily addressed by the authors, who have also done a good job of addressing the caveats associated with extending the findings to other contexts.

Issue 3: Reference to encounter rates as a proxy for abundance: This has been well addressed by the authors.

Issue 4: Choice of analytical model and the need to better account for covariates: The use of the

linear mixed model instead of the two-sided Wilcoxon signed rank tests allows the authors to model the effects of correlated groupings (here concessions, concession pairs and cameras) on encounter rate. Reviewer 1 suggested a hierarchical model structure, for which mixed models are an appropriate choice, however I found it difficult to assess the actual model structure that was used in this study as the text only notes the random effects that were used without detailing the structure of the random effects (presumably cameras were nested within concessions within concession pairs, in a multi-level random effect structure?).

We understand the request of the reviewer to clarify the model structures. Cameras are indeed nested within concessions, and concessions in concession pairs, which is coded in the following way:

```
lmer(log_rai ~ certification * weightclass + (weightclass | pair) + (1 | concession) + (1 | camera), data = weight)
```

The nesting is explicit as each camera and concession has a unique identifier. For further info regarding our way of coding the nesting in lmer we refer to

<https://stats.stackexchange.com/questions/228800/crossed-vs-nested-random-effects-how-do-they-differ-and-how-are-they-specified>

We have now specified the model structure in the text in lines 589 - 592:

“To assess whether encounter rates varied between FSC and non-FSC concessions, we quantified the means of the paired concessions using linear mixed-effects models with concession pairs, concessions and cameras as random effects, whereby cameras were nested within concessions within concession pairs, in a multi-level random effect structure.”

The text also notes that several models were tested, then sorted using BIC—again this is an appropriate choice, but it would be useful to see the model structures that were tested to help the reviewers assess the methods. I note that the code will be made available for interested readers upon publication, however it would be useful if the authors provided a list of the models that were tested as part of the supporting information.

We thank the reviewer for the comment and have now included Table S7 with a list of models that were tested, including structures, degrees of freedom and BIC values.

Lines 828 - 841 of the table text:

“Supplementary Table 5. List of tested models to assess potential covariate influence of geographic and camera trap site covariates. Linear mixed-effects models for mammal encounter rates (‘rai’) were specified with concession pairs (‘pair’), concessions (‘sitename’) and cameras (‘camera’) as random effects, whereby cameras were nested within concessions within concession pairs, in a multi-level random effect structure. Geographic covariates included elevation (‘elevation’), distance to roads (‘dist_roads’), distance to rivers (‘dist_rivers’), distance to settlements (‘dist_settlements’) and distance to protected areas (‘dist_protected_areas’). All geographic covariates were also tested with quadratic terms. Camera trap site covariates included the availability of water within 50 m (‘water’) or fruiting trees within 30 m (‘fruit_trees’), visibility and the presence of trails or paths (‘type_of_site’). Relative importance of models was tested using a model-selection approach based on minimization of Bayesian Information Criterion (BIC) values, reported alongside the number of parameters (npar) and model structures. Models are sorted by ascending BIC in separate analyses for all species combined, for weight classes, taxonomic groups, and IUCN Red List categories.”

We would like to note that we detected a small mistake in how our covariate model structures were set-up. Namely, we did not include concessions as a random factor when testing the influence of covariates. Our reasoning to do so was because geographic covariates were equal per concession (e.g., each concession has unique distance to roads and rivers etc), but in fact, the geographical covariates are quantified per camera (e.g., each camera has unique distance to roads and rivers etc). We therefore decided to reassess the covariate models. Previously, both taxonomic groups and all species combined, indicated that elevation slightly improved the model, but in our reassessment of the covariate models this was not the case anymore for taxonomic groups, and for all species combined, both elevation and distance to rivers slightly improved the model.

We would also like to note that after a careful read of Anderson 2008, we decided to change our interpretation and explanation of the small differences in BIC values between the best models for the analysis whereby all species are combined. Namely, the inclusion of covariates elevation and distance to rivers slightly lowered the BIC value, but the difference with the base model was so low that we decided that evidence for the inclusion of these covariates was negligible. Also see page 89 - 91 of Anderson 2008.

Anderson, D. R. Model based inference in the life sciences: a primer on evidence. vol. 31 (Springer, 2008).

The reviewer's concerns about interactions between covariates appear to have been addressed by the authors' consideration of the quadratic covariate terms.

Issue 5: Accuracy of the hunting sign method: I am not well qualified to answer this concern as monitoring methods to detect hunting presence are outside my expertise, however the author's acknowledgement of the weaknesses of this approach in the manuscript appears to address this issue from my layman's perspective.

Specific comments:

Abstract, line 24: "Over a quarter of the tropical forests is exploited for timber". This is grammatically incorrect—rephrase (possibly "Over a quarter of the world's tropical forests are exploited for timber"?).

We thank the reviewer for the observation and have adapted the sentence accordingly.

Line 25: "Over a quarter of the world's tropical forests are exploited for timber¹."

Line 76 "Hunting not necessarily completely extirpates wildlife species...": Rephrase—e.g. "hunting does not necessarily completely extirpate..."

We thank the reviewer for the observation and have revised the sentence.

Lines 76 - 78: "Hunting does not necessarily completely extirpate wildlife species, especially when forests are connected, but rather results in population declines⁴."

Line 131: "Mammal encounter rates of all IUCN Red List categories were higher"—the p-values in Figure 4 and Extended Data Table 3 suggest that this is not true for Endangered and Vulnerable species. Could the authors explain or rephrase this sentence?

This observation is indeed correct and we have specified which categories were higher in FSC-certified concessions.

Lines 139 - 141: “Mammal encounter rates of IUCN Red List categories Critically Endangered, Near Threatened, and Least Concern, were higher in FSC-certified than in non-FSC concessions (Fig. 4; Extended Data Tables 2 & 3).”

Line 176: “We detected no difference in overall species diversity”. In extended table 4, there are several species that are detected only in either the FSC or non-FSC certified areas (for example there are at least 6 species of primates that were not detected in the FSC-certified areas). Was this an expected outcome, and are there systematic differences in faunal composition that are worth noting here?

We thank the reviewer for the observation, and we recognize that our reference to the difference in species observed in FSC-certified and non-FSC concessions requires more nuance. This is because the species that were observed in only the FSC-certified or the non-FSC concessions were species with very low overall encounter rates. These differences are thus likely pertaining more to chance observations than to management impacts.

We have rephrased the sentence relating to diversity both in the introduction and in the discussion.

Lines 121 - 124: “We observed comparable species diversity in the two concession types, as only a small number of species, all with very low encounter rates, were lacking completely in one or other of the concession types.”

Lines 205 - 207: “We did not observe a loss of species that were encountered frequently in the other concession type, nor did we expect to.”

Line 185 “the loss of large wildlife” (and elsewhere). There are several places where the authors use “wildlife” (e.g. lines 252, 254, 305, 336 etc) or “animals” (line 359, 690), which may be taken out of context since the study only examines mammal species. It would be more precise to use “mammals” everywhere.

We understand the concern and have replaced all instances of the words ‘wildlife’ and ‘animals’ in the context of our methods or results to ‘mammals’.

Line 231: “The FSC approach is holistic and controlling hunting is likely the most important factor for the reduction of environmental impacts in logged tropical forests.” The phrase “The FSC approach is holistic” may be confusing to readers who don’t have the context of the review rebuttal. Consider rephrasing to “FSC certification must satisfy several criteria to support holistic forest management. In logged tropical forests, controlling hunting is likely the most important factor...” (or similar wording—edit as required)

We thank the reviewer for the comment and have revised the sentence.

Lines 221 - 226: “The FSC takes a comprehensive and all-encompassing perspective when it comes to managing and promoting sustainable forest practices. This approach recognizes that forests are complex ecosystems with intricate interconnections between their various components, including flora, fauna, soil, water, and climate. In logged tropical forests, controlling hunting is likely the most important factor for the reduction of environmental impacts in logged tropical forests⁷.”

Lines 253: “the geographic variation... was non-significant”. Is this a statement of statistical significance or a general statement meaning “not important”?

We thank the reviewer for the question, and we have removed the sentence altogether as it was redundant with the text at the start of the paragraph.

Lines 236 - 240: “For the sections of the concessions that we sampled, we ensured comparability between paired concessions. We maximized the similarity in geographic covariates that may drive variation in mammal abundance, i.e., elevation and distances to roads, rivers, human settlements and protected areas, between each pair of FSC-certified and non-FSC concessions (Extended Data Fig. 2; Extended Data Table 4).”

Line 278: “Species home ranges and movement patterns can change in response to disturbance, which can affect encounter rates”. This apparently contradicts with the argument in line 251, i.e. “Slight differences in logging history are not expected to have a large effect on the data because wildlife is mobile and returns quickly to areas that have been exploited”. Am I misinterpreting the latter statement here?

We understand the reviewers’ confusion. The former statement refers to the uncertainty of using encounter rates as an abundance measure, which is now described with more nuance in a following sentence.

Lines 272 - 275: “Species’ home ranges and movement patterns can change in response to disturbance, which can affect encounter rates. It is however unlikely that changes in activity solely make up the observed differences in encounter rates, given the consistency of the data in the three heaviest weight classes.”

Whereas the latter sentence refers to relatively limited temporal impacts of logging.

Lines 322-324: The findings don’t indicate that the FSC regulations lead to more effective control of unsustainable hunting. As noted at line 239, the data in this study don’t allow for causal inference about the specific measures implemented as part of the FSC regulations, so the link to hunting is a hypothesis that is not tested in this study. This sentence should be reworded accordingly.

We thank the reviewer for the sharp observation and have reworded the sentence accordingly.

Lines 325 - 330: “Our findings indicate that the requirements of FSC certification lead to effective mitigation of direct and indirect influences of logging on tropical forest mammals. The control of widespread and unsustainable hunting and poaching that is facilitated by the increased access to forests engendered by timber extraction is probably a key determinant of this impact. However, not all hunting is illegal, and FSC certification protects customary rights to hunt non-protected species for subsistence.”

Note that Table S6 was missing from my review package, so I have not reviewed the material beyond Table S5.

We apologize for the inconvenience, which may have arisen because the table is included as a separate excel file. We will make sure the table is attached to the current version of the manuscript.

Reviewer Reports on the Second Revision:

Referees' comments:

Referee #2 (Remarks to the Author):

The manuscript is certainly better - thanks!

Line 189 careful – given that logging is occurring that will reduce carbon storage whatever large mammals are doing.

201. Need to see the hunting data to support this statement.

259-263 sentence starting Moreover – I don't think that one can make this claim and I would omit.

265 'Although the first priority for species protection should be to maintain unlogged forests where there is effective law enforcement, our results challenge the notion that, at least for large-bodied mammals in WEA, logging is always disastrous'. This is a non-sequeter – the first priority is for unlogged forests does logically lead to logging being disastrous.

272 'This is because wildlife protection measures and law enforcement are applied across all FSC-certified forests, as part of the FSC principles, criteria and indicators' But just because it says that they should do this in principle does not mean they always apply it in practice. The picture being painted here is too rosy. Your next sentence says this – so why raise the issue at all?

288 you have not shown they preserve habitats - so dont say this

293 was there legal hunting in any concessions – if so which? If they were all in non-FSC then we have a problem.

314 English problem

Tim Caro

Referee #3 (Remarks to the Author):

The paper has improved greatly after the authors have made the appropriate corrections. I have made some minor edits on the manuscript which I would like the authors to follow, in particular the use of body mass.

Referee #4 (Remarks to the Author):

I have read the rebuttal letter and manuscript of Zwerts et al. This time I only have two minor comments:

LINE 187: remove "delicate" as it is a qualifier not needed

Line 198: say "sustainable forest management practices" rather than "sustainable forest practices"

Referee #5 (Remarks to the Author):

I am happy with the revisions made in this version of the manuscript.

Referee #6 (Remarks to the Author):

Thank you to the authors for providing comprehensive responses to my requests for further clarification about the mixed model structure. I am satisfied that the random effects model is correctly coded and I appreciate the inclusion of Table S5 so that other readers can examine the full suite of models that were tested. Thank you also for comprehensively addressing my minor comments.

I have no further concerns about this manuscript. Congratulations to the authors on producing a very interesting study.

Response to referees

Referee #2 (Remarks to the Author):

The manuscript is certainly better - thanks!

Line 189 careful – given that logging is occurring that will reduce carbon storage whatever large mammals are doing.

We agree with the referee that logging affects carbon storage. However, since our comparison does not concern unlogged sites, we believe the statement that large mammals are positively correlated to carbon storage is not out of place here.

201. Need to see the hunting data to support this statement.

The statement is backed up by a peer reviewed reference that explicitly mentions the importance of controlling hunting in tropical forests, we therefore believe that providing additional data to back up the statement is not required here.

259-263 sentence starting Moreover – I don't think that one can make this claim and I would omit.

We understand the concern and have omitted the sentence.

265 'Although the first priority for species protection should be to maintain unlogged forests where there is effective law enforcement, our results challenge the notion that, at least for large-bodied mammals in WEA, logging is always disastrous'. This is a non-sequeter – the first priority is for unlogged forests does logically lead to logging being disastrous.

To express more clearly that we refer to the impact of logging on species protection, we have added the words 'for wildlife conservation' at the end sentence:

"Although the first priority for species protection should be to maintain unlogged forests where there is effective law enforcement, our results challenge the notion that, at least for large-bodied mammals in WEA, logging is always disastrous for wildlife conservation."

272 'This is because wildlife protection measures and law enforcement are applied across all FSC-certified forests, as part of the FSC principles, criteria and indicators' But just because it says that they should do this in principle does not mean they always apply it in practice. The picture being painted here is too rosy.

We have good reason to assume that FSC-certified companies put FSC's requirements in practice because FSC-certified companies are audited for compliance, which we have now specified in the sentence:

"This is because wildlife protection measures and law enforcement are applied across all FSC-certified forests, as part of the FSC principles, criteria and indicators for which FSC-certified companies are audited for compliance (Supplementary Tables 1 & 2)."

Your next sentence says this – so why raise the issue at all?

We do however acknowledge that there may be other contexts whereby, despite FSC's standardized regulations, wildlife may be differentially affected than in our study region:

"We infer this with caution as timber extraction volumes, concession size and shape, presence of public roads, population density and other characteristics may differ between concessions and thereby affect the impacts of FSC-certified forest management."

288 you have not shown they preserve habitats - so dont say this

We have indeed not shown that habitats are preserved in our study, although it can be deduced from the wildlife that we observed that habitats are conserved as well. Moreover, it is well known that logged forests preserve habitats much better than agro-industrial alternatives such as oil palm or fibre plantations. As this is also addressed in the cited reference in this sentence we believe that we can maintain the statement.

293 was there legal hunting in any concessions – if so which? If they were all in non-FSC then we have a problem.

Not in the non-FSC concessions. There is legal and regulated hunting in some of the FSC-certified concessions but this generally concerns very low levels of off-take compared to the quantities of illegal hunting in places where hunting controls are not enforced.

314 English problem

Thank you for pointing this out, we have added the words "forest management practices":

"To ensure environmental and socially responsible forest management practices,"

Tim Caro

Referee #3 (Remarks to the Author):

The paper has improved greatly after the authors have made the appropriate corrections. I have made some minor edits on the manuscript which I would like the authors to follow, in particular the use of body mass.

We have adopted almost all minor edits, including the use of body mass instead of weight.

Referee #4 (Remarks to the Author):

I have read the rebuttal letter and manuscript of Zwerts et al. This time I only have two minor comments:

LINE 187: remove "delicate" as it is a qualifier not needed

We removed the word "delicate".

Line 198: say "sustainable forest management practices" rather than "sustainable forest practices"

We have added the word "management" to the sentence.

Referee #5 (Remarks to the Author):

I am happy with the revisions made in this version of the manuscript.

Referee #6 (Remarks to the Author):

Thank you to the authors for providing comprehensive responses to my requests for further clarification about the mixed model structure. I am satisfied that the random effects model is correctly coded and I appreciate the inclusion of Table S5 so that other readers can examine the full suite of models that were tested. Thank you also for comprehensively addressing my minor comments.

I have no further concerns about this manuscript. Congratulations to the authors on producing a very interesting study.

We sincerely thank all referees for their time and effort to improve our manuscript.